# Inhibition of ribosome biogenesis in the epidermis is sufficient to trigger organism-wide growth quiescence independently of nutritional status in *C. elegans*

**Qiuxia Zhao, Rekha Rangan, Shinuo Weng, Cem Özdemir, Elif Sarinay Cenik** [ID] *

Department of Molecular Biosciences, University of Texas at Austin, Austin, Texas, United States of America

* esarinay@utexas.edu

**Data Availability Statement:** The RNA-seq libraries from this study can be accessed via the NCBI GEO database using the accession code

## Abstract

Interorgan communication is crucial for multicellular organismal growth, development, and homeostasis. Cell nonautonomous inhibitory cues, which limit tissue-specific growth alterations, are not well characterized due to cell ablation approach limitations. In this study, we employed the auxin-inducible degradation system in *C. elegans* to temporally and spatially modulate ribosome biogenesis, through depletion of essential factors (RPOA-2, GRWD-1, or TSR-2). Our findings reveal that embryo-wide inhibition of ribosome biogenesis induces a reversible early larval growth quiescence, distinguished by a unique gene expression signature that is different from starvation or dauer stages. When ribosome biogenesis is inhibited in volumetrically similar tissues, including body wall muscle, epidermis, pharynx, intestine, or germ line, it results in proportionally stunted growth across the organism to different degrees. We show that specifically inhibiting ribosome biogenesis in the epidermis is sufficient to trigger an organism-wide growth quiescence. Epidermis-specific ribosome depletion leads to larval growth quiescence at the L3 stage, reduces organism-wide protein synthesis, and induced cell nonautonomous gene expression alterations. Further molecular analysis reveals overexpression of secreted proteins, suggesting an organism-wide regulatory mechanism. We find that UNC-31, a dense-core vesicle (DCV) pathway component, plays a significant role in epidermal ribosome biogenesis-mediated growth quiescence. Our tissue-specific knockdown experiments reveal that the organism-wide growth quiescence induced by epidermal-specific ribosome biogenesis inhibition is suppressed by reducing *unc-31* expression in the epidermis, but not in neurons or body wall muscles. Similarly, IDA-1, a membrane-associated protein of the DCV, is overexpressed, and its knockdown in epidermis suppresses the organism-wide growth quiescence in response to epidermal ribosome biogenesis inhibition. Finally, we observe an overall increase in DCV puncta labeled by IDA-1 when epidermal ribosome biogenesis is inhibited, and these puncta are present in or near epidermal cells. In conclusion, these findings suggest a novel mechanism of nutrition-independent multicellular growth coordination initiated from the epidermis tissue upon ribosome biogenesis inhibition.

GSE213367. The data is available at this link: https://www.ncbi.nlm.nih.gov/geo/query/acc.cgi?acc=GSE213367 Deseq2 analysis output tables for RNAseq, Mass-spec raw peptide counts and DEP output tables are available as supplementary tables, and uploaded to submission.

**Funding:** This work was supported by the National Institutes of Health (5R35GM138340-03 to ESC), the Welch Foundation (F-2133-20230405 to ESC). The funders had no role in study design, data collection and analysis, decision to publish, or preparation of the manuscript.

**Competing interests:** The authors have declared that no competing interests exist.

**Abbreviations:** AID, auxin-inducible degradation; DCV, dense-core vesicle; dsRNA, double-strand RNA; GO, Gene Ontology; GST, glutathione S-transferase; IAA, indole-3-acetic acid; IA-2, insulinoma-associated protein 2; IIS, insulin/insulin-like growth factor signaling; LFQ, label-free quantification; NGM, nematode growth media; RNAi, RNA interference; RNA Pol I, RNA Polymerase I; SEC, self-excising selection cassette; TTR, transthyretin.

## Introduction

Organism-wide growth in metazoans is a complex process that is influenced by a combination of autonomous [1–3] and nonautonomous factors. These factors process information from nutritional cues via pathways including TORC1, TGFβ, and insulin/insulin-like growth factor signaling (IIS) (reviewed in [4,5]). Interestingly, growth coordination maintains proper body proportions, even if a specific organ's growth is hindered. For example, when the left limb of a mouse has its cell cycle suppressed during development, the symmetry between the left and right limb remains unchanged [6]. In *Drosophila*, other compartments' development slows down when one embryonic compartment's growth is disturbed [7–9]. However, how growth regulation occurs in response to a specific organ's growth impairment is not well understood, unlike the mechanisms governing nutrition-dependent organismal growth regulation.

One of the best-studied examples of growth coordination comes from *Drosophila* studies, revealing that the growth of eye discs is coordinated upon knockdown of ribosomal protein genes, *RpL7* or *RpS3*, specifically in the wing tissue [7]. This finding suggests that system-wide growth coordination requires communication between different organs. In *Drosophila*, the coordination between wing and eye disc growth is regulated by the insect-specific Xrp1 and mediated by ecdysone inhibition through the secreted peptide hormone Dilp8. The JNK stress signaling pathway also plays a role in this process [7,10]. Since Xrp1 and Dilp8 are specific to the insect clade, it suggests the existence of evolutionarily divergent mechanisms. However, several key questions remain unanswered: (1) Do similarly divergent or conserved mechanisms operate in other species? (2) What role do specific tissues play in overall organism growth? (3) How is information relayed between body parts?

*Caenorhabditis elegans* provides a suitable model for studying growth coordination due to its fast developmental cycle and available genetic and cytological tools. In contrast to insect clade development, which is centrally mediated by the ecdysone hormone, *C. elegans* developmental timing is dependent on an intricate network of heterochronic genes (reviewed in [11]). Furthermore, *C. elegans* can modulate their larval development according to external cues, such as nutrient availability, through dauer regulation and starvation-induced larval quiescence, primarily attributed to IIS and TGFβ signaling pathways (reviewed in [4,5]). Finally, numerous examples, such as starvation response, dietary restriction, and mitochondrial unfolded protein response-mediated longevity [12–23], demonstrate cell nonautonomous organism-wide communication within *C. elegans*.

Our previous research revealed a ribosome biogenesis-mediated growth coordination in mosaic animals in *C. elegans*. Specifically, using unigametic inheritance [24], we generated embryos with an anterior–posterior (AB-P1) split of wild-type and ribosomal protein gene null cells (*rpl-5(0)*) at the two-cell cleavage step. These mosaic embryos, completing embryogenesis with maternal ribosomes, experienced L1 stage arrest. The growth of wild-type cells paralleled that of their *rpl-5(0)* neighbors, indicating an organism-wide checkpoint. This checkpoint persisted despite insulin signaling pathway bypass mutations (*daf-16* and *daf-18*) and was associated with a stress response gene expression profile, suggesting that growth coordination between the 2 lineages can be independent of nutritional status [25].

In this study, we used an auxin-inducible degradation (AID) system [26,27] to specifically and reversibly modulate ribosome biogenesis at distinct stages in *C. elegans*. Ribosomes, consisting of 2 subunits, 60S and 40S, integrate different ribosomal proteins and ribosomal RNA. The transcription of 45S ribosomal DNA loci into rRNA is carried out by RNA Polymerase I (Pol I) [28,29]. Primarily, in the nucleolus, the newly translated ribosomal proteins are imported from the cytoplasm by dedicated chaperones [30,31]. For instance, Rrb1p chaperones uL3 to the nucleolus, and its depletion reduces the 60S ribosomal subunit levels, leaving

the 40S subunit unaffected in yeast [32]. Similarly, Tsr2 chaperones the r-protein eS26 to the first assembling pre-ribosome, the 90S, and is necessary for the cytoplasmic processing of 20S pre-rRNA into mature 18S rRNA [33]. Tsr2 also regulates the release and reincorporation of eS26 from mature ribosomes, facilitating a reversible stress response [34]. Within *C. elegans*, *rpoa-2* encodes the second largest subunit of RNA Pol I, while *Y54H5A.1 (grwd-1)* and *Y51H4A.15 (tsr-2)* encode the chaperone proteins required for the assembly of ribosomal proteins RPL-3 and RPS-26, respectively.

Using the AID system, we examined the impact of RPOA-2, GRWD-1, and TSR-2 depletion on ribosome biogenesis. We found that depleting any of these proteins led to a deficiency in ribosome biogenesis, triggering a growth quiescence response across the organism at an early larval stage. Interestingly, this quiescence was resistant to rescue attempts by bypass mutations in the insulin signaling pathway (*daf-16* and *daf-18*). The deficiency of ribosome biogenesis in tissues of equivalent volume resulted in a scaled, coordinated growth. We directed our attention towards the specific inhibition of ribosome biogenesis in the epidermis tissue, observing profound consequences for the entire organism. This led to a significant slowdown in organism-wide growth (quiescence) and induced gene expression changes in a diverse range of cell types in a cell nonautonomous manner. Overexpression of secreted proteins and dense-core vesicle (DCV) pathway proteins were observed, while both cytosolic and mitochondrial ribosomal proteins were significantly underexpressed throughout the organism. We also confirmed the overexpression of the DCV membrane-associated protein, IDA-1, in response to epidermal ribosome biogenesis inhibition.

The *ida-1* gene, which exhibits epistasis to *unc-31*, encodes IA-2/IDA-1, a protein that genetically interacts with UNC-31/CAPS and affects neurosecretion in *C. elegans* [35]. UNC-31 is the *C. elegans* homolog of CAPS, a crucial factor in the priming step of $Ca^{2+}$-dependent exocytosis of DCVs and the regulation of DCV cargo release [36,37]. Intriguingly, reducing the expression of epidermal *ida-1* or *unc-31* led to an increase in worm body length when epidermal ribosome biogenesis was inhibited. We also observed the presence of DCV puncta, indicative of the subcellular localization of IDA-1, in or near epidermal cells. Taken together, our findings highlight the significant role of DCV secretion in the vicinity of epidermal tissue in mediating the growth quiescence associated with epidermal ribosome biogenesis inhibition.

## Results

### Modulation of ribosome biogenesis using the AID system

To modulate ribosome biogenesis in an inducible fashion, we decided to use the AID system to target biogenesis factors [26,27]. In this approach, an auxin-inducible degron-tagged target protein can be depleted upon the expression of an auxin receptor F-box protein TIR1 and the small molecule auxin (indole-3-acetic acid (IAA)) [26]. We generated *C. elegans* strains with an AID degron::GFP cassette integrated into the genomic loci of an RNA Pol I subunit (*rpoa-2*), as well as the chaperones of RPL-3 and RPS-26 (*grwd-1/Y54H5A.1* and *tsr-2/Y51H4A.15*, respectively) using CRISPR/Cas9-mediated editing [30,38]. These tagged proteins specifically function in ribosomal RNA transcription from repeated 45S ribosomal DNA loci, as well as nucleolar 40S and 60S ribosome subunit biogenesis, thus, specifically target ribosome biogenesis at 3 distinct steps (**Figs 1A and S1A–S1F**).

To further validate RPOA-2, GRWD-1, and TSR-2 have analogous roles in *C. elegans* ribosome biogenesis as described for their homologs, we conducted polysome profiling experiments. Our data indicated that depleting RPOA-2 reduced the amount of ribosomal subunits, monosome, and polysome peaks, without preferential depletion of a specific subunit (**S2C Fig**). The depletion of GRWD-1 significantly reduced the large subunit (60S), monosome and

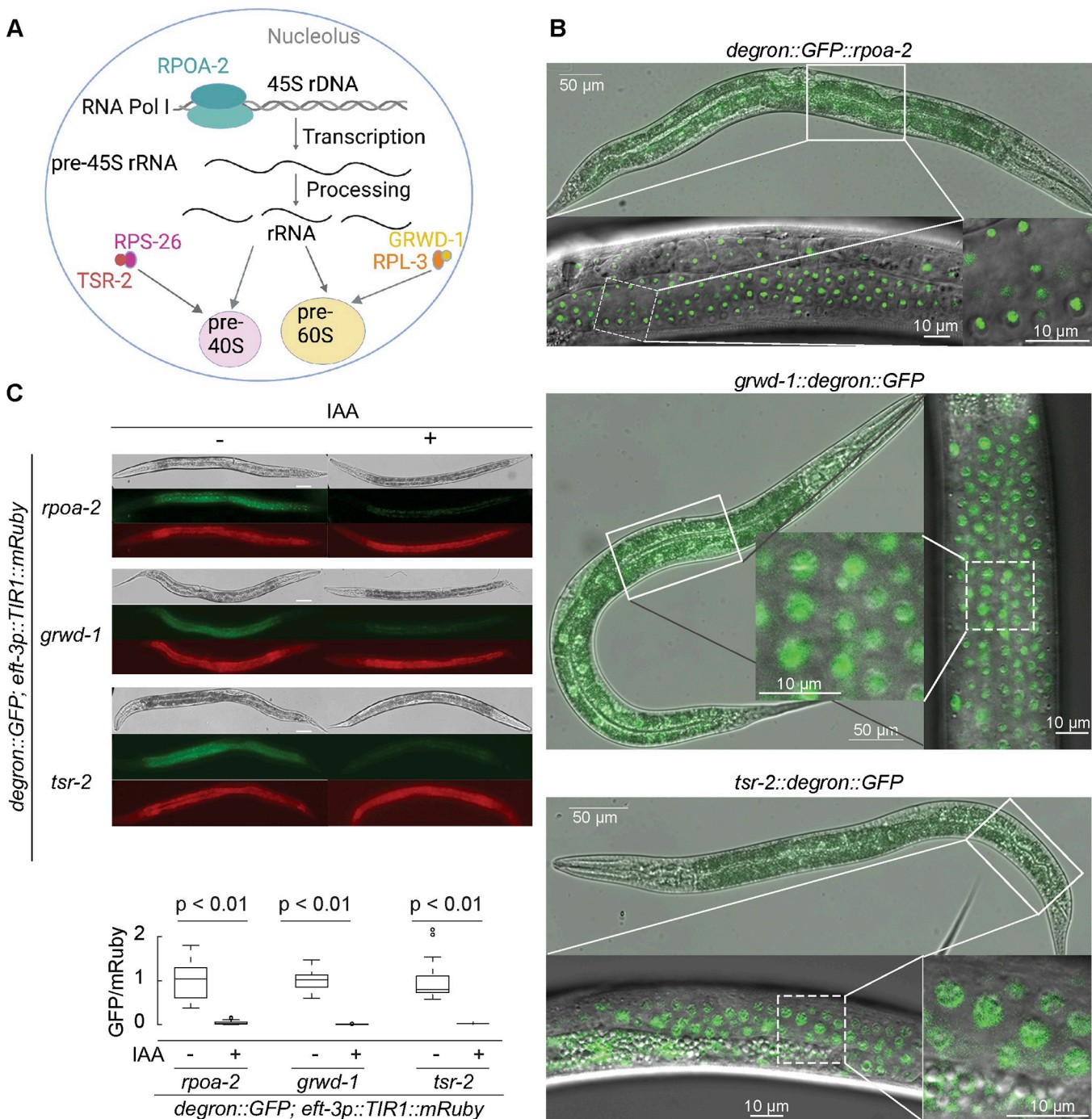

**Fig 1. AID system facilitates degradation of ribosome biogenesis factors.** **(A)** The scheme illustrates ribosome biogenesis factors investigated in this study and is created with BioRender.com. *rpoa-2* encodes the second-largest subunit of RNA Pol I, while *grwd-1* and *tsr-2* encode chaperone proteins that assist RPL-3 and RPS-26 in nuclear large and small ribosomal subunit assembly, respectively. **(B)** Localization of endogenous RPOA-2, GRWD-1, and TSR-2 in live animals. A degron::GFP cassette was integrated to the N terminus of the endogenous *rpoa-2* gene or C terminus of *grwd-1* and *tsr-2* genes. The L4 stage animals were imaged using DIC and fluorescence. RPOA-2 is localized in the nucleolus, while GRWD-1 and TSR-2 are primarily localized in the nucleus. **(C)** The AID system enables the degradation of RPOA-2, GRWD-1, and TSR-2. L3 stage animals were incubated with 1 mM IAA and imaged after 24 hours. For quantification, each 20× image was analyzed using Fiji software. Data represent GFP intensity (corresponding to RPOA-2, GRWD-1 or TSR-2) normalized by mRuby intensity (TIR1) from 25 animals. Animals were immobilized on slides using 1 mM levamisole. Statistical significance was determined via an independent *t* test. Scale bar, 50 μm. The underlying data for (C) can be found in the Tab A in S1 Data. AID, auxin-inducible degradation; IAA, indole-3-acetic acid; RNA Pol I, RNA Polymerase I.

polysome peaks, with an accumulation of the small subunit (40S) (**S2D Fig**). This observation is in line with the previous studies on the yeast ortholog encoded by *RRB1* [32]. TSR-2 depletion led to a decrease in mature ribosomes and an overall increase in 60S levels (**S2E Fig**), in agreement with the earlier studies on the yeast ortholog encoded by *TSR2* [33]. Therefore, our results suggest that depleting RPOA-2, GRWD-1, or TSR-2 significantly reduces translating ribosome populations, a finding that corroborates previous studies on yeast orthologs.

Strains expressing degron::GFP-integrated RPOA-2, GRWD-1, or TSR-2 were found to be homozygous viable and phenotypically identical to the wild type. These exhibited nucleolar RPOA-2 [39], nuclear GRWD-1, and nuclear TSR-2 localization patterns (**Figs 1B and S2A**), indicating that the degron::GFP tags are consistent with normal organism growth. To evaluate the AID system, we crossed strains expressing degron::GFP integrated the ribosome biogenesis factor (RPOA-2, GRWD-1, or TSR-2) with strains ubiquitously expressing TIR1 under the *eft-3* promoter. L3 stage animals expressing both the degron::GFP tag and TIR1 showed complete depletion of GFP signals when exposed to 1mM IAA overnight (**Fig 1C**). Similarly, in the presence of IAA, RPOA-2 tagged with degron::GFP was undetectable by western blot within 3 hours (**S2B Fig**). This suggests that the AID system successfully degraded the ribosome biogenesis factors (RPOA-2, GRWD-1, and TSR-2).

## Embryonic inhibition of ribosome biogenesis results in a reversible quiescence

To assess the effect of IAA-mediated depletion of RPOA-2, GRWD-1, or TSR-2 on embryonic development, we treated stage-synchronized embryos expressing RPOA-2, GRWD-1, or TSR-2 tagged with a degron::GFP, in the presence of ubiquitous TIR1, with 1 mM IAA for 24 hours. As anticipated, given the sufficiency of maternal ribosomes for *C. elegans* embryonic development [25], all embryos completed embryogenesis and hatched, despite the depletion of ribosome biogenesis factors with IAA treatment. To evaluate postembryonic development without new ribosome biogenesis, we measured larval body length following a 3-day incubation (with or without IAA) starting from stage-synchronized embryos (**Fig 2A, top**). All 3 strains (*rpoa-2*, *grwd-1*, *or tsr-2 degron*::*GFP* integrated in the presence of *eft-3p*::*TIR1*) exhibited an overall stall in growth and development when exposed to IAA (**Figs 2A and S3A**).

It is important to note that, in the absence of IAA, the global expression of TIR1 induces a modest background degradation of degron::GFP (**S3B Fig**) [40,41], with a higher basal degradation in *tsr-2*::*degron*::*GFP* strains compared to *rpoa-2* and *grwd-1 degron*::*GFP* strains (**S3B Fig**). Thus, animals ubiquitously expressing TIR1 and degron::GFP-integrated TSR-2 developed significantly more slowly even in the absence of IAA, suggesting that basal degradation of TSR-2 affects postembryonic development (**Fig 2A and 2B**).

To accurately stage animals upon the universal embryonic depletion of RPOA-2, we examined 2 distinct postembryonic lineages: the mesoblast precursor cell (M cell) and vulval precursor cells *hlh-8p*::*GFP* and *egl-17p*::*mCherry* reporters [42,43]. During the L1 stage, the M cell undergoes mitosis to generate 18 cells, 2 of which migrate during the L2 stage, subsequently dividing and differentiating into sex muscle cells at later larval stages [44]. With global depletion of RPOA-2, we observed 18 M cells, indicating that the quiescent larvae progressed at least to the late L1 stage (**Fig 2C, top**). Comparable M cell division patterns in *rpoa-2(ok1970)* null animals (**Fig 2C, bottom**) suggest that the ubiquitous depletion of RPOA-2 by the AID system can mimic the genetic deletion of *rpoa-2*. At the L3 larval stage, vulval precursor cells P (5–7).p adopt primary or secondary cell fates and undergo invariant cell divisions [45]. An inspection of these cells suggests that *rpoa-2(ok1970)* null animals halt development at the L2 stage (**S3C Fig**). In conclusion, the universal depletion of these ribosome biogenesis factors and the genetic loss of *rpoa-2* lead to a growth standstill at the L2 stage.

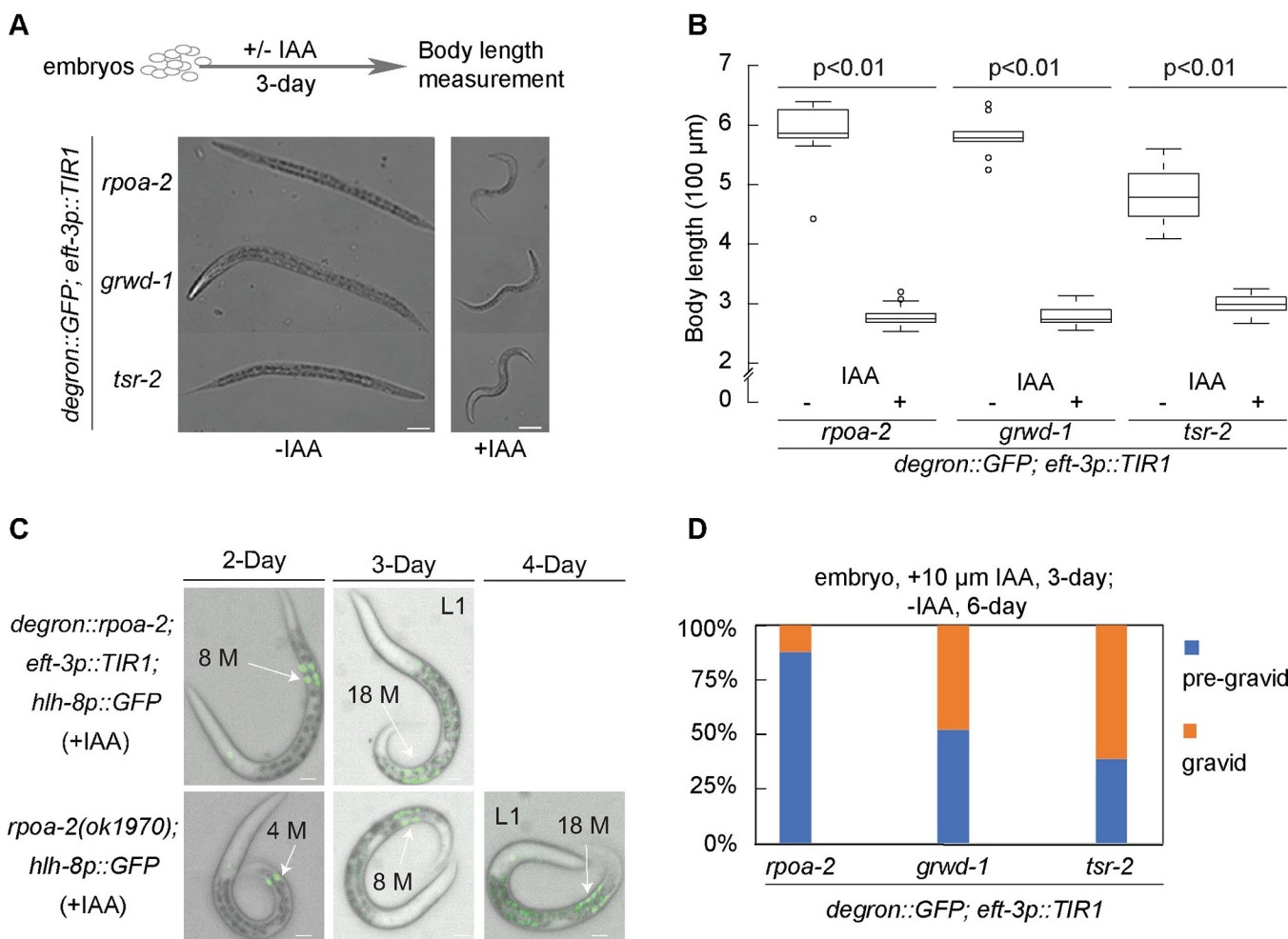

**Fig 2. AID-mediated organism-wide ribosome biogenesis inhibition leads to developmental quiescence at the L2 stage. (A)** Synchronized embryos of *degron*::GFP::*rpoa-2*, *grwd-1*::*degron*::GFP, or *tsr-2*::*degron*::GFP strains in the presence of *eft-3p*::*TIR1* were treated either with (+) or without (−) 1 mM IAA for 3 days. Animals were imaged using DIC. Scale bar, 50 μm. **(B)** The overall body length of animals (from **A**) was analyzed using Fiji software. Data were obtained from 9 animals without IAA treatment and 21 animals with IAA treatment from each strain. Statistical significance was determined using an independent *t* test. **(C)** Mesoblast precursor (M) cell division was observed over a span of 4 days following embryo synchronization. Up to 18 M cells were observed in both *degron*::GFP::*rpoa-2*; *eft-3p*::*TIR1* animals treated with 1 mM IAA (top) and homozygous arrested *rpoa-2(ok1970)* animals (bottom). Scale bar, 10 μm. Animals were immobilized on slides using 1 mM levamisole in (**A** and **C**). **(D)** Synchronized embryos expressing TIR1 globally and harboring degron:: GFP-integrated ribosome biogenesis factors (RPOA-2, GRWD-1, and TSR-2) were incubated with 10 μM IAA for 3 days, followed by 6 days after removal of IAA. The percentage of gravid adults was assessed from at least 40 animals. The underlying data for (**B** and **D**) can be found in Tab B in S1 Data. AID, auxin-inducible degradation; IAA, indole-3-acetic acid.

Contrary to the developmental quiescence observed at the early larval stage, when ribosome biogenesis was inhibited from the L4 stage onward, the animals matured into gravid adults (S4A Fig). This implies that the developmental quiescence is specific to the early larval stage.

During the quiescent larval stage characterized by the depletion of a ribosome biogenesis factor (RPOA-2, GRWD-1, or TSR-2), animals relied on preexisting ribosomes for survival. We then investigated whether these remaining ribosomes could facilitate the recovery of these animals to gravid adulthood by enabling the synthesis of new ribosomes when IAA was removed. AID-mediated protein degradation can be reversed in the presence of low IAA concentrations (10 μM, 25 μM), with a potential for complete protein recovery post IAA removal [26]. To examine this reversibility, embryos were exposed to 10 μM IAA for 3 days and then

transferred to IAA-free plates for 6 days. The recovery rates post IAA removal were notably less than 100% but significantly higher for GRWD-1 and TSR-2 global depletion compared to RPOA-2 (12.2%, 47.6%, and 61%, respectively, after a 3-day depletion of RPOA-2, GRWD-1, or TSR-2) (**Fig 2D**). Additionally, postembryonic growth reversibility for globally depleted RPOA-2 was observed to be both time and IAA concentration dependent, with gravid adults noted after up to 5 days of incubation with 10 μM IAA (**S4B Fig**). These findings suggest that restarting new ribosome biogenesis can alleviate growth quiescence in a fraction of animals.

## Gene expression changes in response to ribosome biogenesis inhibition are distinct from that of L1 starvation and dauer stages

To gain insights into the molecular basis of the reversible larval quiescence caused by ribosome biogenesis inhibition, we carried out an unbiased gene expression analysis. Transcriptome profiles of RPOA-2-depleted L1 animals (*degron::GFP::rpoa-2; eft-3p::TIR1*, +IAA) were compared to controls (*degron::GFP::rpoa-2*, +IAA), revealing 297 genes with significant changes ($p_{adj} < 0.05$) [46] (**Fig 3A and S1 Table**). Overexpressed categories in RPOA-2-depleted larvae included genes related to ribosome maturation, protein synthesis, chromatin and transcription regulation, as well as DNA damage response and repair (**S5A Fig and S1 Table**).

To determine shared and divergent pathways underlying this phenotype, we systematically compared the molecular profiles of RPOA-2 depletion-induced larval quiescence to other conditions of larval arrest. Similarity of gene expression changes was tested between ribosomal protein gene null L1 larvae (*rpl-5* or *rpl-33* null [25]) and RPOA-2-depleted L1 larvae. Significant overlaps were observed between differentially expressed genes in the genetic ribosomal protein null mutants and RPOA-2-depleted animals (**Fig 3B and 3C**). These results suggest that inducible inhibition of ribosome biogenesis elicits a molecular signature akin to the complete loss of ribosome components.

*C. elegans* enters a developmental diapause state in response to post-hatch starvation, a state that can be reversed upon feeding. Furthermore, *C. elegans* can survive adverse environmental conditions by undergoing dauer arrest at the second molt [47]. To better understand the distinct contributions of conditions inducing young larvae quiescence, we compared gene expression changes in animals undergoing RPOA-2 depletion to those of starvation-induced L1 and dauer stages [48,49]. The overexpressed genes shared between starvation-induced L1 or dauer animals and those with RPOA-2 depletion were limited (**Fig 3B, 3D and 3E**, $p = 1$, Fisher's exact test). However, there were significant overlaps among genes underexpressed upon starvation and overexpressed in response to RPOA-2 depletion, and vice versa ($p < 0.01$, odds ratios = 2.9 and 2, respectively). As DAF-16 is activated during starvation [17], we explored if these overlaps represented DAF-16 targets [50]. A similar opposite pattern with DAF-16 targets (ChIP-Seq) under low insulin signaling conditions suggested that DAF-16 is likely not activated during RPOA-2 depletion (**S5B** and **S5C Fig**).

Significant overlaps were observed between genes underexpressed in response to RPOA-2 depletion and those underexpressed during starvation or dauer (**Fig 3D and 3E**, $p < 0.01$, odds ratios = 2.3 and 8.5, respectively, Fisher's exact test). The shared underexpressed genes between dauer and RPOA-2 depletion datasets were significantly enriched for collagen synthesis and cuticle development Gene Ontology (GO) categories ($p < 0.01$, **S2 Table**) and included numerous examples related to molting (for instance, *noah-1*, *noah-2*, *mlt-11*, and *qua-1*). Thus, the shared underexpressed genes in dauer and RPOA-2 depletion might represent genes related to postembryonic development progression.

Given a lack of significant overlap among overexpressed transcripts in response to RPOA-2 depletion and conditions like starvation and dauer, we analyzed similarities with other stress

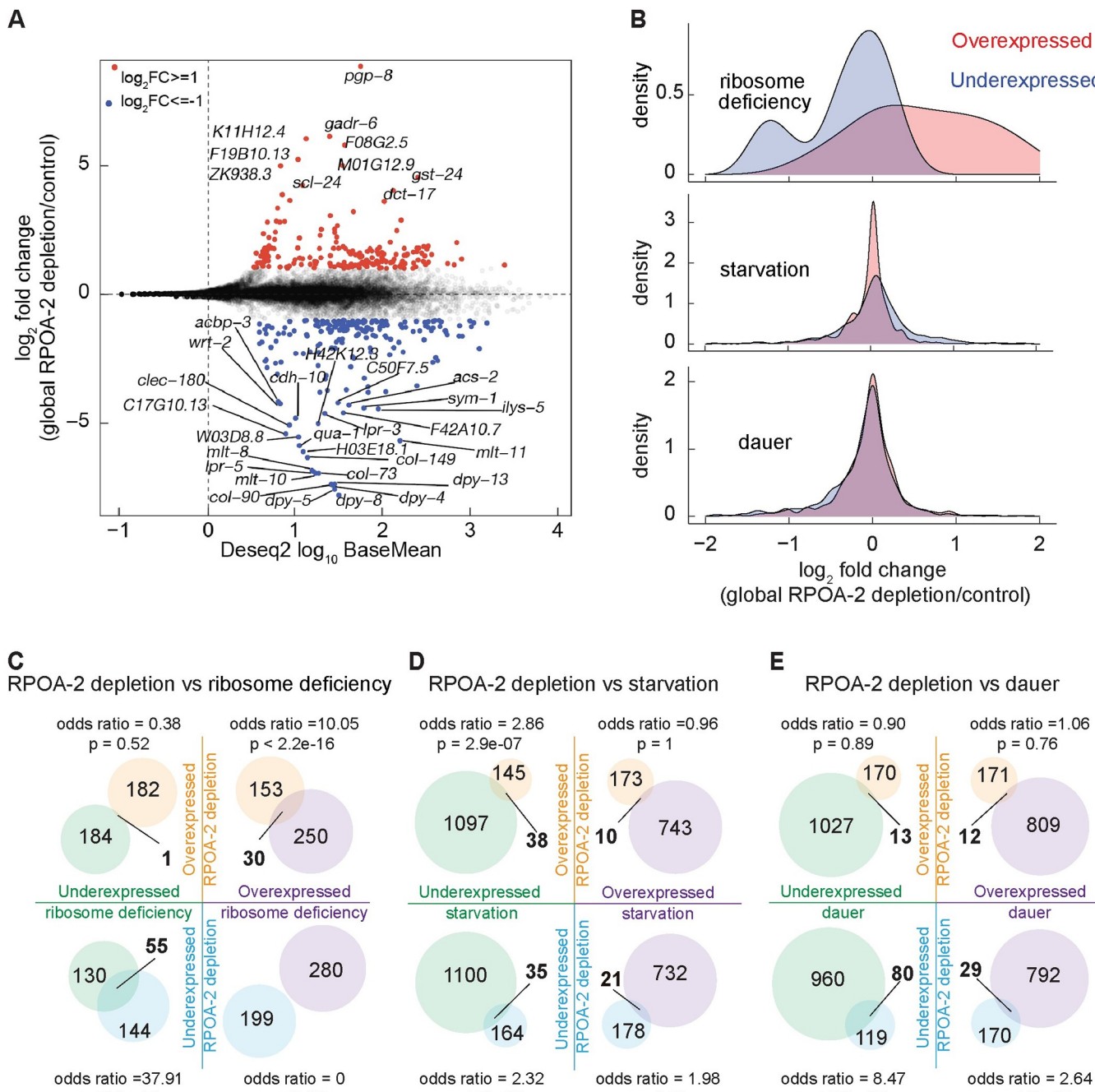

**Fig 3. Gene expression signatures in response to global RPOA-2 depletion. (A)** Log$_2$ fold changes of protein coding genes (y-axis) predicted by Deseq2 analysis of RNA-seq were plotted against predicted base mean values (x-axis). Genes with at least 2-fold significant overexpression and underexpression were marked in red and blue, respectively. Symbols indicate genes that exhibit at least 16-fold differential expression. **(B)** Deseq2 log$_2$ fold change values were represented in a histogram in response to global RPOA-2 depletion compared to overexpressed (light red) and underexpressed (light blue) genes of ribosome deficiency, starvation, and dauer responses. **(C-E)** Shared gene expression changes in response to RPOA-2 depletion by RNA-seq and growth arrest related pathways were illustrated in the Venn diagrams. Significant differentially expressed genes in RPOA-2 depletion animals (change > 2-fold) were compared to that with published data from ribosomal protein null mutants (*rpl-5(0)*, *rpl-33(0)*) [25] **(C)**, starvation-induced L1 [48] **(D),** and dauer animals [49] **(E).** The underlying data for (B-E) can be found in Tab C in S1 Data.

conditions that induce larval growth arrest or diapause. One such condition is UV irradiation, which leads to partial larval arrest. Interestingly, we observed significant overlaps among both over- and underexpressed genes in response to UV irradiation and RPOA-2 depletion (odds ratios = 7.8 and 6.7, respectively, $p < 0.01$, **S5D and S5E Fig**) [51]. We also observed significant overexpression of several DNA damage response genes in RPOA-2-depleted animals (for instance, *rad-50*, *xbp-1*, and *smc-5*, **S1 Table**). These results suggest that RPOA-2 depletion and UV irradiation may activate shared pathways.

## The epidermis-specific inhibition of ribosome biogenesis results in growth quiescence at the L3 stage

Given that the global depletion of ribosome biogenesis factors results in a reversible quiescence, marked by a unique molecular profile, we speculated that this organism-wide response may be triggered by signaling from specific tissues. To investigate this hypothesis, we depleted RPOA-2 in different tissues and assessed the impact of these depletions on overall organism growth.

We assessed organism-wide growth (body length) in animals experiencing global RPOA-2 depletion, comparing them to those where RPOA-2 depletion was tissue specific, using the expression of TIR1 under tissue-specific promoters (*eft-3p* for global, *col-10p* for epidermis [hypodermis], *myo-2p* for pharynx, *ges-1p* for intestine, *myo-3p* for body wall muscle, and *sun-1p* for germ line) [26,27] (**Figs 4A and S6A**). RPOA-2 depletion in different tissues resulted in varying degrees of growth delay, from approximately 33% to approximately 95%, with body proportions being conserved (**S6B Fig**). These results imply that each tissue has a role in orchestrating organism-wide growth.

Strikingly, the epidermis-specific RPOA-2 depletion induced visible growth quiescence (**Fig 4A**). This quiescence was also observed with the epidermal depletion of either GRWD-1 or TSR-2, signifying that inhibiting 40S or 60S subunit biogenesis independently can initiate an epidermis-mediated, dramatic organism-wide growth retardation (**Fig 4B**). After incubation for 3 days with 1 mM IAA from the embryonic stage, animals with epidermis-specific TIR1 and integrated degron::GFP::RPOA-2 remained at an early larval stage (**Fig 4C**). Vulval precursor cell (*egl-17p::mCherry*) examination suggested that these animals developed until the L3 stage under epidermis-specific RPOA-2 depletion (**Fig 4D,** as inspected at 16°C and 20°C). Importantly, IAA removal allowed the reversal of growth quiescence triggered by epidermal degradation of a ribosome biogenesis factor (**S6C Fig**). These findings point to a role for epidermal ribosome biogenesis inhibition in inducing reversible larval growth quiescence.

The insulin IGF-1 signaling from the epidermis can nonautonomously activate P and M lineages at the L1 stage in a *daf-16*-mediated manner [52,53]. We crossed *daf-16(mu86)* and *daf-18(ok480)* to *grwd-1::degron::GFP; col-10p::TIR1* strain to evaluate relative body size or variation in the vulval cell divisions, which typically occur in the early L4 stage [54]. The body length of *daf-18(ok480)*, *daf-16(mu86)*, or *daf-16* and *daf-18* RNA interference (RNAi)-mediated knockdown animals was not significantly increased (**S7A and S7B Fig**). To determine if the *daf-16(mu86)* and *daf-18(ok480)* animals could progress through the early L4 transition, we inspected vulval cell divisions and did not observe any significant differences after 5 days of 1 mM IAA treatment (**S7C Fig**).

In light of the contrasting expression patterns of DAF-16 target genes under low insulin signaling and ribosome biogenesis inhibition at the RNA level (**S5B and S5C Fig**), and the known nuclear localization of the DAF-16 protein under low insulin signaling conditions [55], we aimed to further assess DAF-16 localization. We crossed an endogenously tagged *daf-16* strain (*daf-16::mKate2*) with the inducible epidermal ribosome biogenesis strain (*grwd-1::*

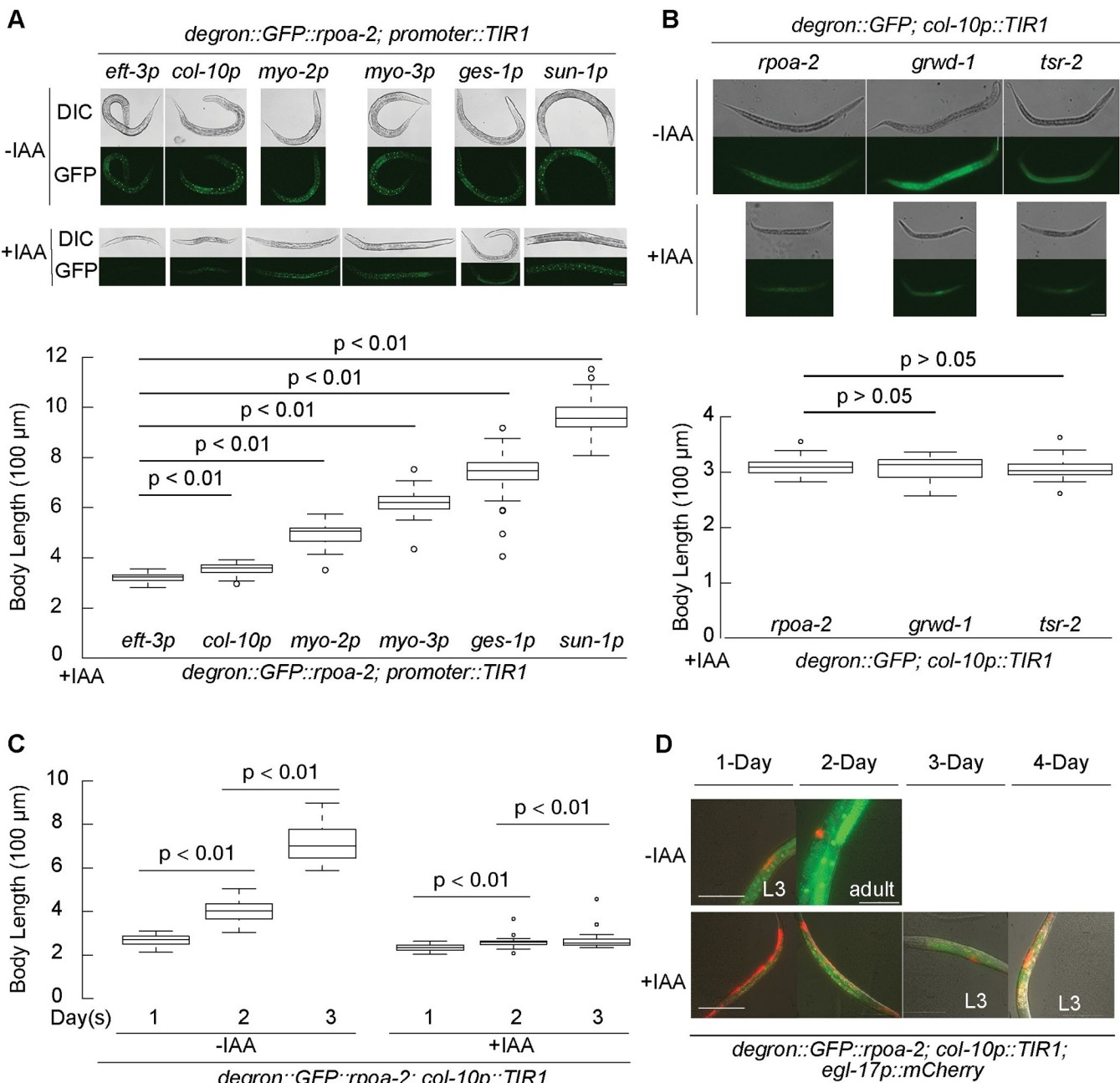

**Fig 4. The epidermis-specific inhibition of ribosome biogenesis results in development quiescence at the L3 stage.** (A) Synchronized embryos of *degron::GFP::rpoa-2* strain, with TIR1 expressed in different tissues driven by the following promoters: global (*eft-3p*), epidermis (*col-10p*), pharynx (*myo-2p*), body wall muscle (*myo-3p*), intestine (*ges-1p*), and germ line (*sun-1p*) were treated with 1 mM IAA for 3 days. GFP signals display the RPOA-2 expression pattern. Body length of animals with RPOA-2 depletion in different tissues was analyzed by Fiji software from 50 animals. (B) Synchronized embryos of strains expressing degron::GFP-integrated ribosome biogenesis factors (RPOA-2, GRWD-1, or TSR-2) and TIR1 in the epidermis were treated with (+) and without (−) 1 mM IAA for 3 days. The body length post-IAA treatment was analyzed from 20 animals using Fiji software. Epidermis (*col-10p*)-specific degradation of GRWD-1 or TSR-2 results in a body length similar to that of RPOA-2. (C) Embryos of the *degron::GFP::rpoa-2; col-10p::TIR1* strain were treated with (+) or without (−) 1 mM IAA, and body length was measured over the span of 3 days from 40 animals for each condition. *P* values were calculated using independent *t* test and adjusted with Bonferroni correction in (A-C). (D) The vulva invariant cell lineage marker (red) (*egl-17p::mCherry*) expression patterns in the strain of *degron::GFP::rpoa-2; col-10p::TIR1; egl-17p::mCherry* under normal conditions (−IAA) and upon epidermis-specific RPOA-2 depletion (+IAA). mCherry appeared from 3-day and 4-day incubations in epidermal RPOA-2-depleted animals, suggesting an L3 stage growth quiescence. Animals were imaged over the span of 4 days from L1. Scale bars in (A, B, D), 50 μm. All animals were immobilized on slides using 1 mM levamisole. The underlying data for (A-C) can be found in Tab D in S1 Data.

*degron*::*GFP; col-10p*::*TIR1*). Under normal conditions, we were unable to detect nuclear localization of DAF-16::mKate2 in L1 larvae either with (−IAA) or without (+IAA) epidermal ribosome biogenesis as expected (**S7D Fig**). However, when these animals were subjected to starvation, DAF-16::mKate2 was observed to localize in the nucleus (**S7E Fig**). Intriguingly, nuclear DAF-16::mKate2 density decreased when epidermal ribosome biogenesis was inhibited (+IAA), in comparison to the control (−IAA) during starvation (**S7E Fig**). This observation suggests that the activation of DAF-16 might be suppressed during epidermal ribosome biogenesis inhibition. From these findings, we conclude that the L3 quiescence observed in response to the epidermis-specific inhibition of ribosome biogenesis is likely independent of insulin IGF-1 signaling.

## Global and epidermal inhibition of ribosome biogenesis results in a shared gene expression program

Observing that epidermis-specific depletion of RPOA-2 results in larval quiescence, we next sought to understand if similar or different gene sets were differentially expressed in this condition compared to a global depletion of RPOA-2. Epidermal RPOA-2 depletion led to 538 differentially expressed genes (**S8A Fig** and **S1 Table**, $p_{adj} < 0.05$). Remarkably similar gene expression profiles were observed between global and epidermis-specific ribosome biogenesis inhibition, sharing 154 overexpressed and 136 underexpressed targets (**Figs 5A, 5B** and **S8B** and **S1 Table**). This suggests a common gene expression response to both global and epidermis-specific RPOA-2 depletion.

To gain further insight into the significantly over- and underexpressed genes, we identified enriched (GO terms among shared targets of global and epidermis-specific RPOA-2 depletion [56]. Overexpressed genes were notably enriched in 2 GO categories: nuclear pore/nuclear part and proteasome. Underexpressed genes revealed significant GO term enrichment for cuticle formation and molting-related categories. Additionally, transmembrane transporter activity was enriched among the combined over- and underexpressed genes (**Fig 5C** and **S1 Table**). The genes involved in cuticle formation and molting facilitate growth or developmental specialization in *C. elegans*, which aligns with the observed phenotypes of animals depleted of RPOA-2 either globally or epidermally.

If an organism-wide response to ribosome biogenesis inhibition occurs, we would anticipate seeing gene expression changes in nonepidermal tissues even when RPOA-2 is specifically depleted in the epidermis. Using previously published single-cell RNA-seq data from L2 animals [57], we identified numerous genes that, although not expressed in the epidermis, displayed differential expression when ribosome biogenesis was perturbed in the epidermis (**Figs 5D, S9A and S9B**). For instance, *phat-5*, a transcript expressed in the pharyngeal gland, muscle, and cholinergic neurons, was underexpressed in animals with epidermal RPOA-2 depletion (**Fig 5D**). Similarly, *app-1*, which was overexpressed in response to epidermal RPOA-2 depletion, encodes an X-prolyl aminopeptidase and is expressed in neurons and the germ line [58] (**S9B Fig**). These findings underscore that an epidermis-specific disturbance of ribosome biogenesis can elicit gene expression changes in both epidermal and nonepidermal cells, including neuronal and intestinal cells. Thus, epidermal-specific manipulations can indeed drive gene expression changes indicative of interorgan communication.

## Organism-wide proteome sculpting responds to epidermis-specific ribosome biogenesis inhibition

The ribosome biogenesis factors RPOA-2, GRWD-1, and TSR-2 are integral in maintaining ribosome concentrations. However, changes in RNA abundance might not fully reflect

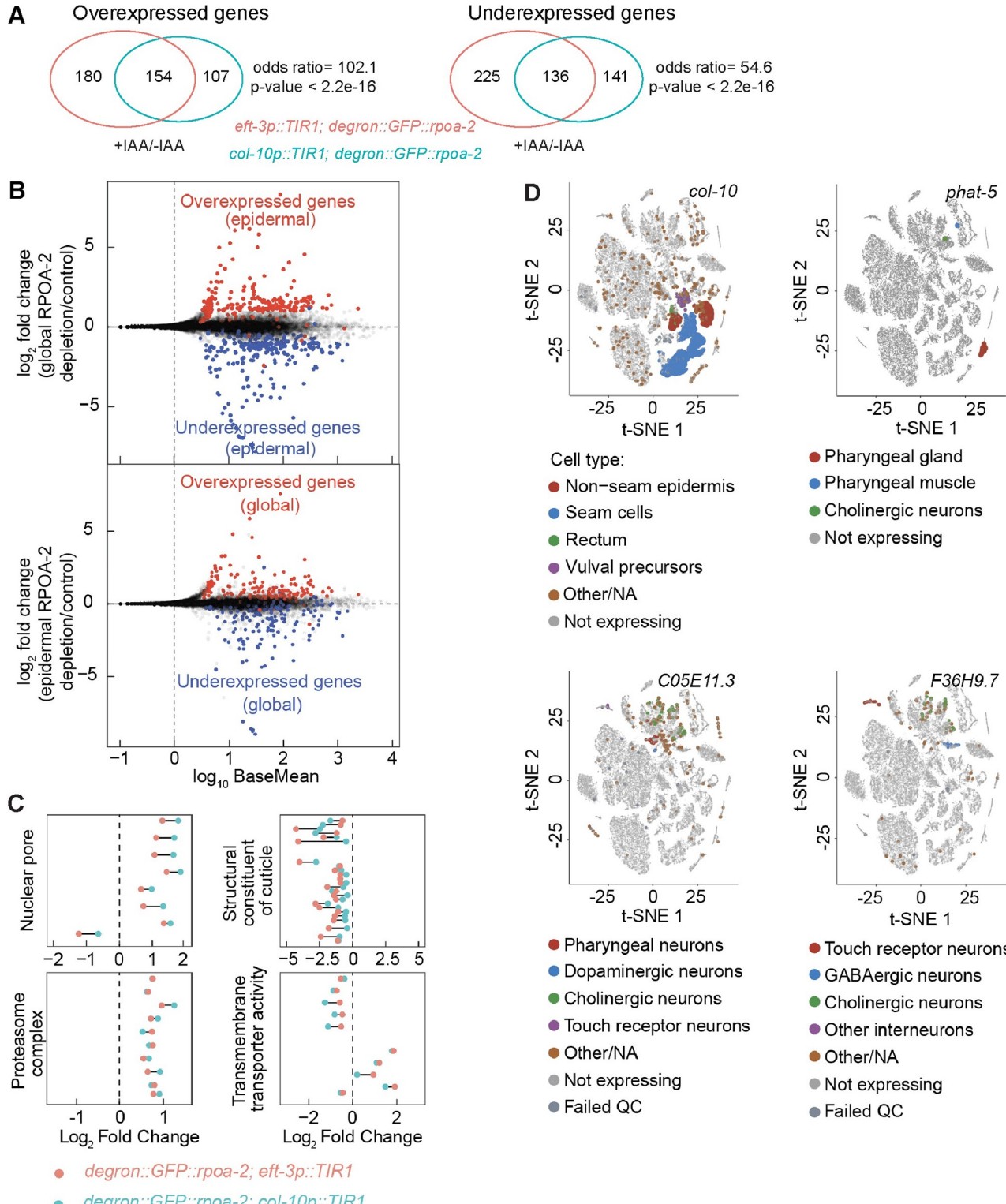

**Fig 5. Inhibition of ribosome biogenesis in epidermis results in a gene expression signature similar to the global inhibition, with notable nonepidermal gene expression changes. (A)** Analysis of RNA-seq data for differentially expressed genes in response to epidermal and global RPOA-2 depletion. The number of shared overexpressed and underexpressed genes was illustrated in the Venn diagrams. **(B)** Log$_2$ fold change of genes (y-axis) from Deseq2 analysis of epidermal RPOA-2 depletion (*degron::GFP::rpoa-2; col-10p::TIR1*, top) and global depletion of RPOA-2 (*degron::GFP::rpoa-2; eft-3p::TIR1*, bottom) compared to the control (*degron::GFP::rpoa-2*) were plotted with base mean values (x-axis). Genes that were more than 2-fold

overexpressed or underexpressed in response to epidermal depletion of RPOA-2 were colored (red and blue, respectively) on the gene expression data for global depletion of RPOA-2 (top). Similarly, genes that were more than 2-fold differentially expressed in response to global depletion of RPOA-2 were colored on the gene expression data for epidermal RPOA-2 depletion (bottom). **(C)** Gene annotation (GO) enrichment categories of shared significantly changed genes in response to both global and epidermal RPOA-2 depletion were performed using Funcassociate 3.0 [56]. The log$_2$ fold changes of these genes in 4 unique GO categories were plotted in response to epidermal (light blue) and global (red) depletion of RPOA-2. **(D)** Single-cell t-SNE plots for representative underexpressed genes (*col-10, phat-5, F36H9.7, C05E11.3*) in response to epidermal RPOA-2 depletion were generated using single-cell RNA-seq data from L2 stage animals [57]. The location of colored points from t-SNE plots are original; however, their size was enlarged to ease visualization. Animals for RNA-seq were grown on 1 mM IAA NGM plates from embryos. RPOA-2-depleted animals were incubated for 24 hours, and control animals for 18 hours. The underlying data for (C, D) can be found in Tab E in S1 Data. GO, Gene Ontology; IAA, indole-3-acetic acid; NGM, nematode growth media.

changes in protein translation or degradation. Specifically, significant reductions in protein synthesis were observed in both global and epidermis-specific ribosome biogenesis inhibition as measured by the heat shock–inducible expression of a fluorescent protein mKate2 (*hsp-16.41p::mKate2; grwd-1::degron::GFP; eft-3p::TIR1* and *grwd-1::degron::GFP; col-10p::TIR1*) (**Fig 6A**). Despite this substantial decrease in new protein synthesis, it persisted during the epidermal depletion of ribosome biogenesis (**Fig 6A**).

To further investigate how the epidermal ribosome biogenesis inhibition affects protein synthesis in other tissues, we utilized label-free intensity-based mass spectrometry to study the proteomic changes following epidermis-specific RPOA-2 depletion. We detected 258 proteins differentially expressed in epidermal RPOA-2-depleted animals compared to controls (**S10A Fig** and **S3 Table**). Interestingly, we observed a significant overlap between genes differentially expressed at both the RNA and protein levels. Additionally, we discovered genes with significant changes at the protein level that were undetectable by RNA-seq (**Fig 6B and 6C** and **S3 Table**). The cellular location and function of these genes are summarized (**S10B Fig**).

To uncover the wider organismal responses at the translational level upon RPOA-2 depletion in the epidermis, we identified enriched GO terms using Funcassociate 3.0 [56]. We found that overexpressed proteins were significantly enriched for the GO terms extracellular space and oxidation–reduction (**Fig 6D** and **S3 Table**). We were particularly intrigued by the enhanced levels of secreted proteins, notably those detected in the epidermis according to single-cell RNA expression data [57]. This observation led us to investigate the potential role of epidermally expressed secreted proteins in controlling body size during epidermal RPOA-2 depletion.

We observed that nearly the entire family of transthyretin (TTR) proteins were overexpressed in response to epidermal RPOA-2 depletion (**Fig 6E**). *C. elegans* TTR proteins are usually secreted and are involved in a wide range of processes (example: TTR-33 is protective against oxidative stress [59]. Glutathione S-transferases (GSTs) [60], which belong to the oxidation–reduction category, were found to be specifically overexpressed at the protein level (**Fig 6E**). GST proteins catalyze the conjugation of reduced glutathione to xenobiotic compounds for detoxification. For example, GST-24 was overexpressed by 2-fold in response to epidermal RPOA-2. Overexpression of GST-24 is linked to enhanced oxidative stress resistance, whereas depleting GST-24 by RNAi leads to reduced stress resistance [61].

Underexpressed proteins were predominantly associated with collagen and cuticle development, biosynthetic processes, and isomerase activity (**Fig 6D** and **S3 Table**). Interestingly, we noticed specific alterations at the protein level for cytoplasmic and mitochondrial ribosomes. Despite their RNA expression remaining stable, their protein abundances markedly decreased during epidermis-specific RPOA-2 depletion (**Fig 6F** and **S10C**).

Considering that the epidermis accounts for about one-seventh of all cells in *C. elegans*, these findings likely reflect changes in other tissues in addition to the epidermis. This comprehensive view of the candidate proteins, their function, and response to RPOA-2 depletion in

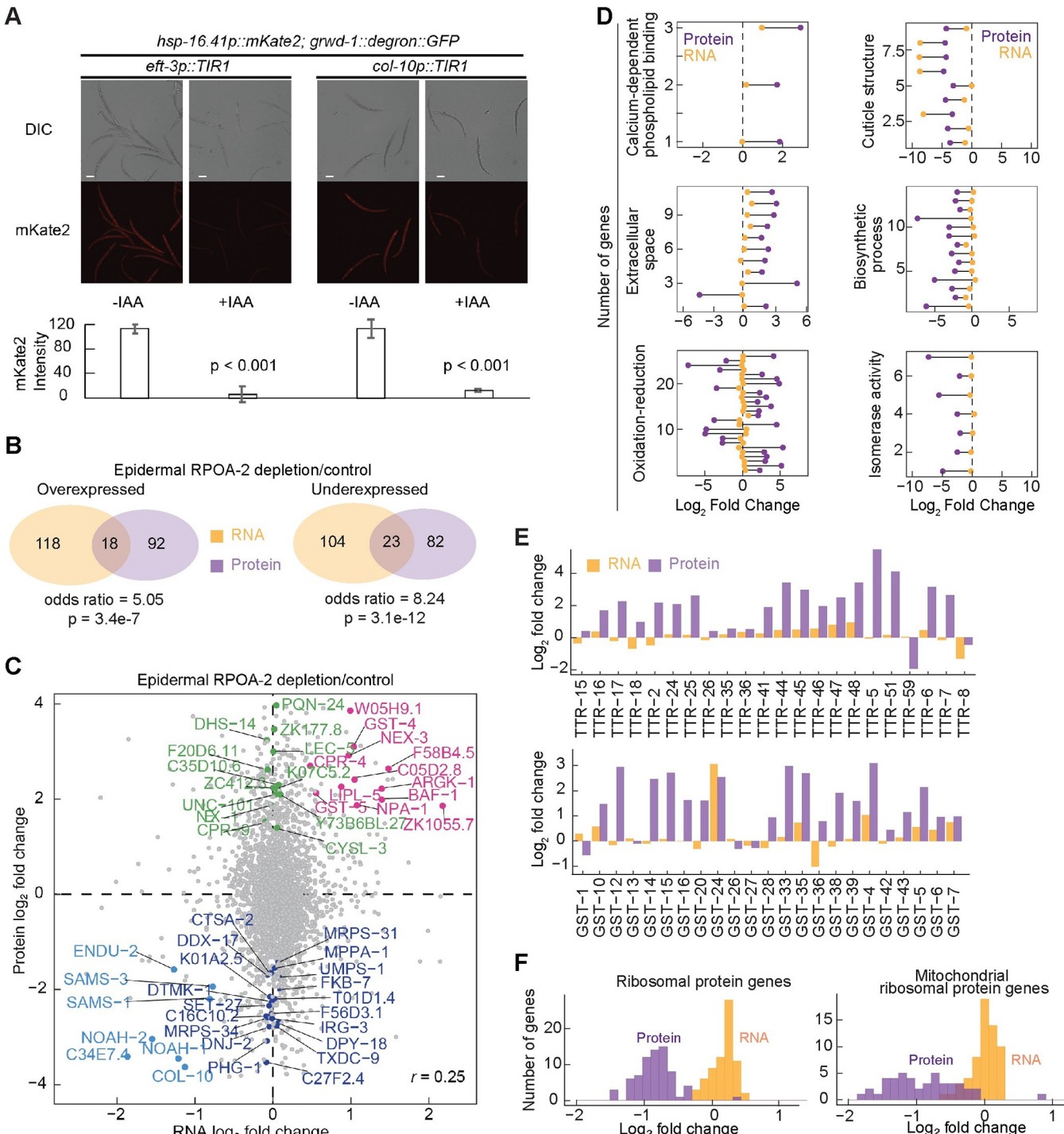

**Fig 6. Global protein-level changes in response to epidermal ribosome biogenesis inhibition. (A)** Representative images of transgenic strains (*hsp-16.41p*::*mKate2; grwd-1*::*degron*::*GFP; eft-3p*::*TIR1* and *hsp-16.41p*::*mKate2; grwd-1*::*degron*::*GFP; col-10p*::*TIR1*) that were grown from embryos on NGM with and without 1 mM IAA for 24 hours at 20°C and then were exposed to heat shock (34°C for 3 hours) before imaging. *hsp-16.41p*::*mKate2* was inducibly expressed by heat shock in both stains. When new ribosome biogenesis was inhibited globally (*eft-3p*::*TIR1*, +IAA), mKate2 protein synthesis was dramatically reduced. Epidermal ribosome biogenesis inhibition (*col-10p*::*TIR1*, +*IAA*) also resulted in a global reduction of mKate2 protein synthesis. For quantification, each 20× image was analyzed using Fiji software. Data are expressed as mKate2 mean pixel density obtained from at least 3 independent experiments with at least 20 animals for each. Animals were immobilized on slides using 20 mM sodium azide. Statistical significance was determined using an independent *t* test. Scale bar, 50 μm. **(B)** Shared gene expression changes by analysis of RNA-seq and label-free proteomics data in response to RPOA-2 depletion were illustrated in the Venn diagrams. Gene expression changes detected by RNA-seq were denoted in orange and proteomics data in purple. **(C)** Log$_2$ fold changes of protein (y-

axis) and RNA (x-axis) levels were plotted (Pearson correlation ($r$) = 0.25). Genes that were significantly differentially over- or underexpressed at both the protein and RNA levels were labeled in orange and light blue, respectively. Genes that are robustly expressed at the protein level but remain unchanged at the RNA level (at least 20 counts of raw reads in any of the replicates and with a ratio of approximately 1) were labeled in green and purple, respectively. **(D)** Significant GO enriched categories were detected by significantly differentially expressed proteins [56]. Six representative significant GO categories with respective protein log$_2$ fold changes were plotted. Each point represents a single protein; orange and purple indicate gene expression changes at the RNA and protein levels, respectively. Genes with the raw RNA-seq counts less than 20 cpm were removed for robust RNA detection and assessment of protein-level changes in (C, D). **(E)** The expression of TTR and GST family genes at RNA (orange) and protein (purple) levels was plotted in bar charts. **(F)** Expression of cytoplasmic (left) and mitochondrial (right) ribosomal protein genes in response to epidermal ribosome biogenesis inhibition was plotted in histograms where orange and purple indicate log$_2$ fold changes at the RNA and protein levels, respectively. The underlying data for (A, and D-F) can be found in Tab F in S1 Data. GO, Gene Ontology; GST, glutathione S-transferase; IAA, indole-3-acetic acid; NGM, nematode growth media; TTR, transthyretin.

the epidermis offers a deeper understanding of the cellular and organismal response to changes in protein synthesis.

## Involvement of *unc-31* in the epidermal ribosome biogenesis inhibition mediated development quiescence

Given the global changes in gene expression observed when epidermal ribosome biogenesis is inhibited, and the altered expression of secreted proteins, we hypothesized that vesicle-mediated transport may be important for the transport of hormones or other molecules. This would allow for communication between organs and coordination of a global growth slowdown. This hypothesis is supported by our preliminary RNAi screen data where the gene *unc-31*, the *C. elegans* equivalent of CAPS (a key component in Ca$^{2+}$-dependent exocytosis of DCVs and regulation of cargo release) [36,37], emerged as a significant suppressor of ribosome deficiency-induced larval quiescence (**Fig 7A**).

Larvae with a null mutation in the ribosomal protein gene, *rps-23(0)*, are arrested at the L1 stage with M cell division in a fraction of animals [25]. However, upon treating these *rps-23(0)* larvae with *unc-31* RNAi, we observed significantly more larvae with divided M cells. This suggests that *unc-31* may have a role in overcoming the larval arrest phenotype (**Fig 7A**). To further explore the potential of *unc-31* in suppression of the growth quiescence induced by the inhibition of epidermal ribosome biogenesis, we simultaneously knocked down the *unc-31* gene expression through RNAi and inhibited epidermis-specific ribosome biogenesis.

Knocking down *unc-31* under standard conditions did not significantly affect animal growth. However, when UNC-31 reduction was combined with the depletion of RPOA-2 in the epidermis, we observed a significant increase in animal size (**Fig 7B**). This finding suggests that a signal, perhaps excreted in response to the disruption in ribosome synthesis, could inhibit organism growth and development during larval quiescence.

To further detect the UNC-31 role in suppression of the growth quiescence, we introduced a null mutant, *unc-31(e928)*, to the inducible epidermal ribosome biogenesis inhibition strain [62]. We observed that *unc-31(e928)* mutants were significantly smaller when the epidermal ribosome biogenesis was inhibited (**Fig 7C**). These findings led us to hypothesize that UNC-31's role in neuromuscular control of foraging behaviors, expressed in neurons and body wall muscles, might be essential for enabling adequate feeding and consequently promoting growth. To investigate this, we generated RNAi strains specific for neurons and body wall muscles. We achieved this by crossing specific mutants with strains, in which *sid-1* or *rde-1* genes were expressed in neurons or body muscle cells, within a background that allows for inducible inhibition of epidermal ribosome biogenesis. RNAi knockdown of *unc-31* in both of these contexts led to smaller animals, substantiating the role of both neuronal and muscle-specific UNC-31 in indirectly promoting growth (**Fig 7D and 7E**).

Although UNC-31 is predominantly expressed in neurons, recent single-cell RNA-seq studies have revealed its expression in nonneuronal tissues including epidermis (**S11A Fig**; [57]).

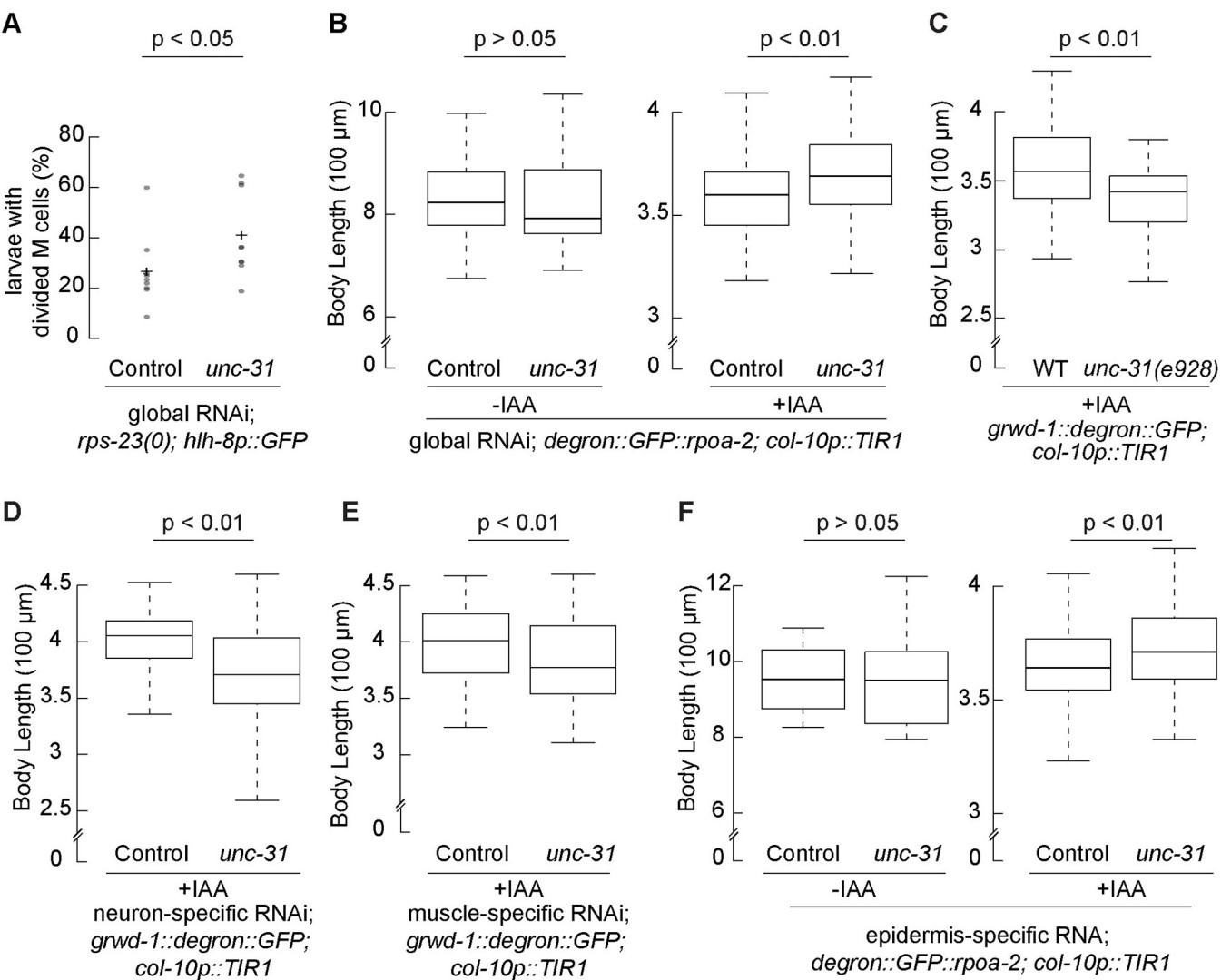

**Fig 7. Epidermal UNC-31 is involved in the epidermal ribosome biogenesis inhibition-mediated growth quiescence.** (A) Synchronized embryos of the strain with a ribosomal protein gene null, *rps-23(0)*, and M cell lineage marker (*hlh-8p::GFP*) were grown on the plates seeded with RNAi bacteria targeting the *unc-31* gene or control for 3 days. Larvae with divided M cells were assessed. Reducing *unc-31* expression by RNAi increased the percentage of larvae with divided M cells. Data were obtained from 9 independent experiments with at least 15 animals for each. Statistical significance was determined using an independent *t* test. (B) Embryos expressing degron::GFP-integrated RPOA-2 and TIR1 in the epidermis were treated with and without 1 mM IAA and fed by RNAi bacteria targeting *unc-31* gene or control for 3 days. Without IAA treatment, RNAi *unc-31* did not affect worm body length (left). With IAA treatment, animals fed by *unc-31* RNAi bacteria grew larger compared to control. (C) A null allele of *unc-31* mutant, *unc-31(e928)*, was crossed to an inducible epidermal ribosome biogenesis strain (*grwd-1::degron::GFP; col-10p::TIR1*). *unc-31(e928)* mutants grew smaller compared to wild type, when epidermal GRWD-1 was depleted (+IAA). (D-F) Tissue-specific RNAi strains were crossed with an inducible epidermal ribosome biogenesis inhibition strain to detect the function of UNC-31 in different tissues. (D) Neuron-specific *unc-31* RNAi animals (*grwd-1::degron::GFP; col-10p::TIR1; sid-1(pk3321); unc-119p::sid-1*) grew smaller without epidermal new ribosomes (+IAA). (E) Reducing *unc-31* expression in muscle (*grwd-1::degron::GFP; col-10p::TIR1; rde-1(ne300); myo-3p::rde-1*) reduced body length with epidermal ribosome biogenesis inhibition (+IAA). (F) Reducing epidermal *unc-31* expression (*degron::GFP:: rpoa-2; col-10p::TIR1; rde-1(ne219); wrt-2p::rde-1*, -IAA) did not change worm body length (left). When the epidermal ribosome biogenesis was inhibited (+IAA), animals with reduced *unc-31* expression in epidermis resulted in a larger body length. Synchronized embryos were incubated for 3 days. Animals were immobilized using 0.5% 1-phenoxy-2-propanol. Each 5× image was analyzed by a custom MATLAB script (S1 Text). Data with IAA treatment were obtained from 3 independent experiments with at least 18 animals for each; data without IAA treatment were analyzed from 26 animals. Statistical significance was determined using an independent *t* test. The underlying data for (A-F) can be found in Tab G in S1 Data. IAA, indole-3-acetic acid; RNAi, RNA interference; WT wild type.

This led us to further investigate whether epidermal UNC-31 suppresses growth when epidermal ribosome biogenesis is prevented. Upon inducing *unc-31* knockdown specifically in the epidermis, we observed an increase in organism-wide growth during the inhibition of epidermal ribosome biogenesis (**Fig 7F**). This result was further validated by additional RNAi experiments using injection and soaking methods, along with feeding (**S11B and S11C Fig**), suggesting a growth-suppressive role for epidermal UNC-31. In summary, expression of UNC-31 in neurons and body wall muscles appears to promote growth, while epidermal UNC-31 seems to act oppositely, negatively regulating organism-wide growth during periods of epidermis-specific ribosome biogenesis disruption.

Interestingly, we observed elevated levels of additional DCV membrane proteins IDA-1/IA-2 and RAB-3 following epidermis-specific ribosome biogenesis inhibition (**S3 Table**). This was intriguing because, despite *unc-31* being predominantly expressed in the nervous system, it suggested a possible role of DCVs in the epidermis. To test this hypothesis, we first generated a reporter strain to examine the presence of *rab-3* transcript in epidermal cells. The cells were labeled with nuclear and cytoplasmic markers driven by the *col-10* promoter. We found that the *rab-3* transcript colocalized with these markers, indicating its expression in the epidermis (**S12A and S12B Fig**).

To further elucidate the factors contributing to growth quiescence, we investigated the potential involvement of the DCV pathway by targeting 4 DCV components: *ida-1*, *rab-3*, *ric-19*, and *unc-108*. Among these, *ida-1* knockdown notably increased worm body length when ribosome biogenesis was inhibited in the epidermis, while reductions in the other genes had no apparent effect (**Fig 8A**, $p_{adj} < 0.05$, *t* test). The *ida-1* gene encodes a protein, known as insulinoma-associated protein 2 (IA-2), which is part of DCVs and interacts with UNC-31/CAPS, impacting neurosecretion in *C. elegans* [35–37]. Furthermore, IDA-1 was also overexpressed when ribosome biogenesis in the epidermis was inhibited (**S3 Table**). To further explore this, we generated a reporter knock-in at the C-terminus of endogenous *ida-1* gene. This allowed us to visualize IDA-1, which is an integral component of DCVs. We identified DCV puncta in or near the epidermal cells (**Figs 8B and S12C**) and detected a significant increase in IDA-1 expression when the epidermal ribosome biogenesis was inhibited (**Fig 8C**), further supporting our hypothesis. These results suggested a crucial role of DCV secretion in mediating organism-wide quiescence response upon interruption of epidermis-specific ribosome synthesis.

Given the suggested role of DCV secretion, we hypothesized that epidermally localized secreted proteins overexpressed in response to this interruption might impact growth. To investigate this, we considered 9 overexpressed, secreted, and epidermally localized candidates (CPR-4, LBP-1, LBP-2, FAR-1, NPA-1, CPR-1, MEC-5, TTR-5, and TTR-2). We reduced the expression of each specifically in the epidermis and observed that 4 of them (*lbp-2*, *far-1*, *cpr-1*, and *ttr-2*) resulted in statistically significant differences in body size when their expression was reduced in the presence of epidermis-specific ribosome biogenesis inhibition (**Fig 8D**, $p_{adj} < 0.05$, *t* test).

Among the investigated candidates, FAR-1 stood out due to its extent of its impact on body size, comparable to UNC-31. The FAR-1 protein belongs to the FAR family, which are small, helix-rich, and secreted proteins that bind to fatty acids and retinol [63–66]. These proteins are linked to numerous biological functions, such as development, reproduction, host infection, and disruption of plant defenses [63–66]. In nematodes, the *far-1* mRNA is specifically expressed in the epidermis [65–67], possibly secreted playing a role in regulating response of nematodes to their external environment [66]. The epidermal expression of *far-1* and its subsequent reduction in epidermis resulting in a significant increase in body length suggests its potential role as a growth inhibitor during epidermal ribosome biogenesis inhibition.

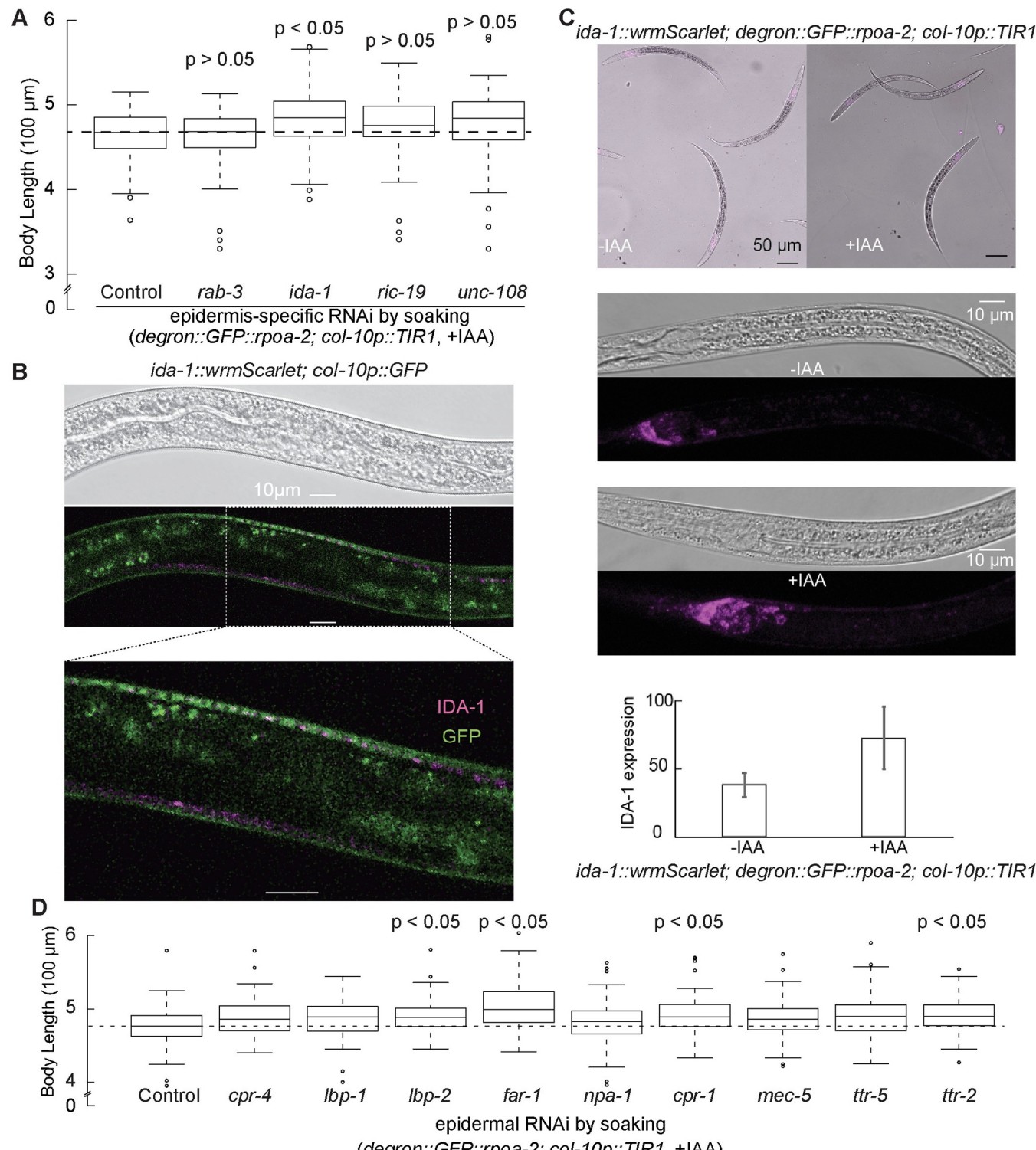

**Fig 8. DCV pathway may be involved in epidermal ribosome-mediated growth quiescence.** (**A**) Using epidermis-specific RNAi strain, 4 DCV pathway components (*rab-3*, *ida-1*, *ric-19*, and *unc-108*) were tested for worm growth regulation in response to epidermal ribosome biogenesis inhibition. Reducing the expression of *ida-1* increased worm body length in the absence of new epidermal ribosomes, while reducing the expression of other components did not affect worm growth. Data were obtained from 3 independent experiments with at least 20 animals for each. *P* values were calculated by independent *t* test and adjusted by Bonferroni correction. The black dash line on the plot indicates the median body length of the control group. (**B**) Localization of endogenous IDA-1 and epidermal cells in live animals. A fluorescent protein gene, *wrmScarlet*, was inserted in the C terminus of the endogenous *ida-1* gene. The magenta color

shows the IDA-1 expression pattern, while green indicates the epidermal cells, marked by *col-10* promoter. L3 to L4 stage animals were immobilized using 1 mM levamisole. **(C)** The wrmScalet-tagged IDA-1 strain was crossed with the inducible epidermis-specific ribosome biogenesis inhibition strain. Synchronized embryos of the strain were grown on NGM plates with and without 1 mM IAA for 24 hours. Animals were immobilized by 20 mM sodium azide. For quantification, each 63× image was analyzed using Fiji software Z project. **(D)** Using the epidermis-specific RNAi strain, secreted proteins that were overexpressed in epidermal RPOA-2 depletion were tested for worm growth regulation in response to epidermal ribosome biogenesis inhibition (*degron::GFP:: rpoa-2; col-10p::TIR1*, +IAA). Data were obtained from 3 independent experiments with at least 20 animals for each replicate. *P* values were calculated by an independent *t* test and adjusted by Bonferroni correction. The black dash line on the plot indicates the median body length of the control. The underlying data for (**A**, **C**, **D**) can be found in Tab H in S1 Data. DCV, dense-core vesicle; IAA, indole-3-acetic acid; NGM, nematode growth media; RNAi, RNA interference.

## Discussion

In this study, we utilized the AID system to modulate ribosome biogenesis at varied stages, thus achieving spatial and temporal resolution. The interference in ribosome biogenesis factors (RPOA-2, GRWD-1, and TSR-2) led to significant reductions in ribosomal RNA transcription and the biogenesis of small and large ribosomal subunits. This approach allowed us to conduct a precise, tissue-specific analysis of ribosome biogenesis within a metazoan system.

When ribosome biogenesis was globally inhibited, we observed a reversible organism-wide postembryonic quiescence at the L2 larval stage. This quiescence presented a distinctive gene expression signature, reminiscent of an activated stress response. Interestingly, this quiescent state was reversible for up to 5 days. This indicated that intact ribosome pools were sufficient to restart the synthesis of new ribosome components and thus reprogram the development of the quiescent L2 larvae into fertile adults. Given that the relative half-lives of metazoan ribosomes are about 5 days [68], we expect that approximately half of the ribosomes would degrade under ribosome biogenesis inhibition. The remaining half is likely to be sufficient to reignite the growth process and restore fertility in adults. This suggests that a longer ribosome life span can be advantageous under conditions that cause quiescence, such as the dauer stage, to reinitiate normal growth once conditions return to normal.

To investigate the mechanism of ribosome biogenesis inhibition-mediated larval quiescence, we compared the resulting changes in RNA abundance to a broad range of gene expression signatures associated with growth quiescence. Interestingly, RPOA-2 depletion led to a gene expression response that was distinct from dauer and starvation-induced quiescence, with no significant enrichment among overexpressed genes.

Given that nutrition-mediated quiescence phenotypes in *C. elegans* are known to be cell nonautonomous [16,17], we aimed to determine whether the ribosome biogenesis-mediated quiescence was similarly triggered from a specific tissue. We conducted experiments to inhibit ribosome biogenesis in volumetrically equivalent tissues [69]. Surprisingly, we observed a significant impact on overall organism-wide growth across all tested tissues, while maintaining relatively conserved body proportions. These findings suggest that different tissues within an organism can coordinate organism-wide growth to varying degrees. Understanding the mechanisms underlying interorgan communication in this context presents an intriguing avenue for further research.

Even though all examined tissues had significant impact on growth, epidermis-specific depletion of ribosome biogenesis had the most prominent impact. Interestingly, among all the tested tissues, inhibition of ribosome biogenesis only in the epidermis resulted in reversible quiescence at the L3 stage. Consistent with these results, when a ribosomal protein gene (*rps-11*) or a translation initiation factor (*egl-45*) was specifically knocked down in the epidermis of *C. elegans*, a noticeable growth impairment was observed. This impairment was accompanied by increased $H_2O_2$ production, enhanced thermal resistance, alterations in AMP/ATP and ADP/ATP ratios, and reduced pharyngeal pumping [70]. These observations collectively suggest the presence of an organism-wide nonautonomous response.

We propose two possible explanations for these findings. First, the inhibition of ribosome biogenesis in the epidermis may impair the synthesis of rate-limiting factors necessary for organism-wide growth. Epidermis may play a crucial role in producing factors essential for overall growth, and their deficiency could impact the entire organism's growth. For example, *C. elegans* cuticle collagens, which are synthesized from the epidermis tissue, can act as regulators of body size through feedback regulation of BMP signaling [71,72]. Second, another possible explanation is that the inhibition of ribosome biogenesis in the epidermal tissue triggers an active stress response, which then propagates throughout the organism. Hence, epidermis might serve as an initiator of a stress response that stunts growth in other tissues and organs.

These two hypotheses have distinct predictions. In the first scenario, where rate-limiting factors are responsible for determining organism-wide growth, the preexisting ribosomes may still be capable of synthesizing the required amounts of such factors over an extended period. As a result, a slower larval progression and growth would be anticipated. Conversely, if the inhibition of ribosome biogenesis in the epidermis triggers an active organism-wide stress response, we would expect to observe significant similarities in gene expression patterns to those seen with global ribosome biogenesis inhibition. These widespread changes would likely manifest in other specialized cell types as well.

In previously reported examples of growth coordination in *Drosophila*, a synchronized developmental delay within the eye disc was observed in response to the wing-specific knockdown of *RpL7* and *RpS3* [7,10]. We found that the AID system is more robust for depleting ribosome biogenesis compared to RNAi. Consequently, the resulting quiescence phenotype in response to epidermis-specific ribosome biogenesis inhibition could lead to a significant slowdown in growth, rather than a mere delay. This distinction could be explained by the fundamental differences in the methodologies used.

Remarkably, our findings show a substantial overlap in the gene expression responses between global and epidermis-specific ribosome biogenesis inhibition (**Fig 5A**). Furthermore, we observed both underexpression and overexpression of cell type–specific transcripts, which are normally undetectable in the epidermis tissue, upon epidermis-specific depletion of RPOA-2. These observations suggest that the L3 quiescence state likely represents an active organism-wide response. Despite the similarities in the gene expression profiles between global and epidermis-specific RPOA-2 depletion, there is a notable distinction in the growth outcomes. Animals with epidermis-specific RPOA-2 depletion can still grow until the L3 stage, whereas animals with global RPOA-2 depletion only reach the L2 stage. These findings suggest that other tissues besides the epidermis contribute to the organism-wide growth quiescence response triggered by ribosome biogenesis inhibition.

Why does the epidermis have a more pronounced impact on overall growth compared to other equally vital tissues, such as the pharynx or intestine? One possible explanation lies in the unique role of the epidermis as the outermost layer exposed to the external environment. The epidermis-mediated quiescence response may play a crucial role in promoting survival in the presence of unexpected stressors, such as UV irradiation or toxins released by pathogenic bacteria. Thus, the epidermis, with its direct exposure to external stressors, may have evolved mechanisms to exert a more significant influence on overall growth and development in response to ribosome biogenesis inhibition, prioritizing the organism's ability to withstand external challenges.

Proteomic analyses provided valuable insights into the changes occurring throughout the organism in response to epidermis-specific ribosome biogenesis inhibition. Here, we identified genes with altered protein abundance despite no apparent changes in RNA expression. We identified numerous overexpressed secreted and extracellular proteins that will provide a basis for future studies on the mechanisms of organism-wide growth coordination. Several

overexpressed proteins stood out, specifically those that are also expressed and secreted in the epidermis. Among these candidates, we discovered that when *far-1* was knocked down in the epidermis, it considerably mitigated the growth stagnation due to epidermis-specific ribosome biogenesis inhibition. The potential role of FAR-1 in fatty acid and retinoid transport suggests its significance in physiology, considering the well-established role of retinoids in growth, development, and differentiation. However, the relatively modest effect sizes observed imply that the compensatory mechanisms could potentially buffer the impact of knockdown of a single gene.

Analysis into the effects of epidermal ribosome biogenesis inhibition revealed a significant role for the epidermally expressed *unc-31* gene (ortholog of CAPS). This gene plays a part in mitigating growth quiescence, suggesting a potential role for DCV secretion from the epidermis. While the expression of numerous neuropeptides in the epidermis is well documented (reviewed in [73]), our study provides evidence demonstrating the involvement of the *unc-31* in the context of the epidermis in *C. elegans*. However, it is important to note that *unc-31* may contribute to nonautonomous growth coordination through its role in additional tissues that are not tested. Similarly, the promoters we utilized for tissue-specific targeting might not be exclusively specific. As such, the effects we detected could be a consequence of changes not only in the intended tissues but also in unintended ones. For instance, UNC-31's operation in neurons found beneath the epidermal cells may be an additional contributing factor to the overall growth inhibition we observed. Finally, the modest effects could be attributable to an inefficiency in RNAi knockdown under the conditions we utilized, the redundant functions of multiple communication pathways or the involvement of other mechanisms, such as physical pressure or membrane contacts.

When we examined other components of the DCV pathway, 2 of which are overexpressed at the protein level, we found that IDA-1, a homolog of human PTPRN2, also significantly mitigates the organism-wide growth quiescence induced by epidermal ribosome biogenesis inhibition. Interestingly, IDA-1 is expressed in epidermal cells according to single-cell RNA-seq datasets, and we observed IDA-1 tagged vesicle puncta in or near epidermal cells. These results overall suggest that the epidermis or neurons near the epidermal cells might be playing a role in coordinating organism-wide growth in response to the epidermal ribosome biogenesis inhibition.

## Materials and methods

### Generation of strains

Constructs and worm strains used in this study are listed in **S4** and **S5 Tables**. All *degron-GFP-c1^sec^3xflag*-tagged gene constructs with a self-excising selection cassette (SEC) were generated using Gibson assembly and verified by sequencing of new junction regions [38]. A codon-optimized *degron* sequence was assembled from gBlocks (IDT) (AF-ESC-702) (**S6 Table**). This coding sequence was used to insert the N-terminus of GFP in pDD282 containing *GFP-c1^sec^3xflag_ccdb*. In the resulting construct pQZ38, *degron* and *GFP* are separated by a Gly-Ser-Gly sequence linker.

The *degron*::*GFP* tagged *rpoa-2* allele was constructed using Cas9 protein driven by *eft-3* promoter in pDD162 and gRNA targeting a genomic sequence in the N-terminus of *rpoa-2* in pRR13, a derivative of pRB1017, an empty vector for gRNA cloning. The sgRNA construct pRR13 was generated by the oligos ESC-RR-5 and ESC-RR-6. All the oligos used in this study are listed in **S6 Table**. *degron-GFP-c1^sec^3xflag* repair template (pQZ43) was constructed for generating the knock-in into the N terminus of the *rpoa-2* gene. The 5′ and 3′ homology arms were amplified 751 bp upstream of *rpoa-2* start codon using oligos ESC-RR-1 and ESC-QZ-

143, and 566 bp downstream of start codon using ESC-RR-3 and ESC-RR-4. The repair templates were used to replace the ccdB in pQZ38.

The *degron*::*GFP*-tagged *grwd-1* gene allele was constructed in a similar manner as above using pDD162 and gRNA targeting a genomic sequence in the C-terminus of *grwd-1* in pQZ73. Oligos ESC-QZ-266 and ESC-QZ-267 were used to anneal sgRNA. The following reagents were used to assemble the final repair template pQZ83: 5′ homology arm (744 bp upstream of *grwd-1* stop codon), 3′ homology arm (947 bp downstream of *grwd-1* stop codon) were amplified using oligos ESC-QZ-270, ESC-QZ-271, ESC-QZ-272, and ESC-QZ-273.

Oligos ESC-QZ-233 and ESC-QZ-234 were used to generate sgRNA targeting C-terminus of *tsr-2* gene in pQZ66. 5′ homology arm (588 bp upstream of *tsr-2* stop codon), 3′ homology arm (654 bp downstream of *tsr-2* stop codon) were amplified using oligos ESC-QZ-237, ESC-QZ-238, ESC-QZ-239, and ESC-QZ-240 to replace the ccdB of pQZ38 as the repair template pQZ69.

The *wrmScarlet*-tagged *ida-1* gene was constructed using a similar manner as above using pDD162 and gRNA targeting a genomic sequence in the C-terminus of *ida-1* in pQZ92. Oligos ESC-QZ-374 and ESC-QZ-375 were used to anneal sgRNA. Oligos ESC-QZ-378, ESC-QZ-379, ESC-QZ-380, and ESC-QZ-381 were used to amplify 5′ and 3′ homology arms of *ida-1* and replace the ccdB of pGLOW39 generating the construct pQZ94.

The inducibly expressed reporter *hsp-16.41p*::*mKate2* was integrated to the loci of *ttTi5605* by the gRNA and cas9 expressed from pDD162. *hsp-16.41* promoter amplified from the plasmid pAP087 replaced the ccdB of pAP087 *ttTi5605 SEC ccdB^2x mKate2^PH^3xHA* generating pQZ89.

All plasmids for microinjection were purified using the Invitrogen PureLink HiPure Plasmid Miniprep Kit (K210002). Oligo sequences used to generate these plasmids are in **S6 Table**.

N2 animals were injected with a mix consisting of 50 ng/μl pDD162 (Cas9 vector), 50 ng/μl gRNA pRR13, 50 ng/μl *rpoa-2* repair template of pQZ43, 5 ng/μl extrachromosomal marker pCFJ104 to produce ESC318. ESC405/406 and ESC402/403/404 were generated by injection a mix containing 50 ng/μl pDD162 (Cas9 vector), 50 ng/μl gRNA pQZ66, 50 ng/μl *tsr-2* repair template pQZ69, 5 ng/μl extrachromosomal marker pCFJ104 and a mix containing 50 ng/μl pDD162 (Cas9 vector), 50 ng/μl gRNA pQZ73, 50 ng/μl *grwd-1* repair template pQZ83, 5 ng/μl extrachromosomal marker pCFJ104 to N2 animals. ESC711 was generated by injection a mix containing of 50 ng/μl pDD162 (Cas9 vector), 50 ng/μl gRNA pQZ92, 50 ng/μl *ida-1* repair template of pQZ94, 5 ng/μl extrachromosomal marker L3785 to N2 animals. ESC351 strain was injected with a mix of 50 ng/μl pDD122 (Cas9 vector and gRNA), 50 ng/μl repair template pQZ89 for integration of *hsp-16.41p*::*mKate2*. Each knock-in was isolated as previously described [38]. The SEC was then excised by heat shock to produce ESC319, ESC424/430, ESC431/432, ESC716, and ESC717.

Strains expressing TIR1 in particular tissues were crossed to *degron*::*GFP*-tagged *rpoa-2* strain to generate strains expressing both degron fused RPOA-2 and TIR1. Strains with global (*eft-3p*) and epidermal (*col-10p*) expression of TIR1 were also crossed with *degron*::*GFP*-inserted *tsr-2* and *grwd-1* strains.

We also crossed strains with tissue-specific RNAi, fluorescent reporters with strains with the AID system. We generated a neuron-specific RNAi strain within an inducible epidermal ribosome biogenesis inhibition background. This was achieved by crossing the *sid-1(pk3321)* mutant with neuronally expressed *sid-1 (unc-119p)*, alongside *grwd-1*::*degron*::*GFP*; *col-10p*::*TIR1*. For the body wall muscle-specific RNAi strain, we crossed *rde-1(ne219)* with body wall muscle-specific *rde-1 (myo-3p)*, again in a context that allows inducible inhibition of epidermis-specific ribosome biogenesis *(grwd-1*::*degron*::*GFP; col-10p*::*TIR1)*. For an epidermis-

specific RNAi effect, we crossed *rde-1(ne219)* with an epidermis-specific *rde-1 (wrt-2p)* in a background that enables inducible inhibition of epidermis-specific ribosome biogenesis (*degron*::GFP::*rpoa-2; col-10p*::TIR1).

CA1210, DV3800, CA1199, HAL230, PD2638, PD2632, VC2372, PD4666, FX30167, RDV55, CB928, PD2635, PD2643, QK52, TU3401, WM118, DV3799, OH13908, WBM1144, HS445, and DLW109 were purchased from CGC. Details are provided in **S5 Table.**

## Worm growth

*C. elegans* strains were grown at 16˚C or 20˚C on agar plates containing nematode growth media (NGM) seeded with *Escherichia coli* strain OP-50 for maintenance culture. Animals in **Figs 1**, **2**, **4A–4C**, **S2A, S2B**, **S3A, S3B**, **S4**, **and S6** were grown at 16˚C, and **Figs 4D**, **6A**, **7**, **8**, **S2C–S2E**, **S3C**, **S7**, **S11**, **and S12** at 20˚C. To obtain synchronized embryos, adult animals were bleached using a buffer containing 0.5 N NaOH and 1.25% sodium hypochlorite for 6.5 minutes. Bleached embryos were placed onto NGM plates. *tsr-2*::degron::GFP; *eft-3p*::TIR1 strain grew on NGM seeded with *E. coli* HB101.

## Auxin (IAA) treatment

The natural auxin IAA was purchased from Alfa Aesar (#A10556). A 400-mM stock solution in ethanol was prepared and stored at −20˚C. IAA was diluted into the NGM agar and cooled to about 55˚C before pouring plates. Plates were left at room temperature for 1 to 2 days to allow bacterial lawn growth. Controls for experiments using IAA are NGM plates with an equivalent concentration of ethanol.

## Polysome fractionation

L3 larvae grown on regular NGM plates were transferred to NGM plates with and without 1 mM IAA for 24 hours at 20˚C. Animals were liquid nitrogen flash frozen in polysome lysis buffer [74] and ground in liquid nitrogen (with mortar and pestle). The frozen worm powder was thawed on ice and solubilized in polysome lysis buffer that was supplemented with 1 mM DTT, 100 μg/ml cycloheximide, 40 U/100 μl recombinant ribonuclease inhibitor (Invitrogen), 2 U/100 μl DNase (Invitrogen). Lysates were loaded onto 10% to 50% sucrose gradients and spun for 2.5 hours at 40,000 rpm using SW 40 Ti rotor in an ultracentrifugation system (Beckman Coulter). RNA from monosome and polysome peaks was isolated using a density fractionation system (Brandel). The data were used for the analysis in **S2C**, **S2D and S2E Fig**.

## Worm body length analysis

Worm morphological comparisons were imaged at 5× or 20× magnification with a DIC filter (Leica Imager). Worm body length comparisons were made in Fiji using the segmented line tool down the midline of each animal from head to tail. We developed a worm body length analysis toolbox supported by MATLAB, which could automatically measure worm body length. The script is attached in **S1 Text**. **Figs 7** and **S11** were analyzed by MATLAB. For multiple comparisons of worm body length, we applied the Bonferroni correction by multiplying the *P* value derived from the *t* test by the number of comparisons tested.

## Reversibility assay

Synchronized embryos were placed onto NGM plates with varying concentrations of IAA and number days of treatment, as indicated in **Figs 2D** and **S4B**, and then the IAA-treated larvae were transferred to fresh NGM plates without IAA to observe the phenotypes.

## RNA interference by feeding and injection

*E. coli* strain HT115 (DE3) containing L4440, which expresses double-stranded RNA of a specific sequence fragment, was utilized for RNAi against a gene of interest or a nontarget sequence that does not target the *C. elegans* genome (control). To prepare the bacteria, HT115 was cultured overnight (for 6 to 18 hours) in LB medium supplemented with 50 µg/ml ampicillin at 37˚C. Subsequently, the bacteria were spread onto NGM plates containing 1 mM isopropyl-β-D-thiogalactoside (IPTG) and 25 µg/ml carbenicillin. The plates were incubated overnight to allow the bacteria to generate double-strand RNA (dsRNA). Synchronized embryos were then placed on these plates and allowed to grow for 3 days at 20˚C.

RNAi was also performed by injection, targeting the *unc-31* gene in the epidermis tissue. For this, we combined a 0.8-kb fragment of the *wrt-2* promoter from the *C. elegans* genome, a 451-bp fragment of *unc-31* from cDNA, and a control gene (*mKate2*) from the plasmid pAP087. Additionally, a hygromycin gene with a global promoter *rps-0* was used. PCR-fusions were created by amplifying the *wrt-2* promoter with both the sense and antisense *unc-31* fragments, using the *wrt-2* promoter fragment and *unc-31* fragment mixture as templates. The resulting constructions targeting epidermal *unc-31*, along with the control gene and hygromycin resistant gene, were injected at a concentration of 100 ng/µl into N2 animals [75]. Transgenic lines were screened using a concentration of 250 µg/ml hygromycin.

## M cell division assessment

L4 stage *rps-23[cc5995, A67X]/tmc20* animals were fed with *unc-31* RNAi bacteria to evaluate the division of a single M cell in the F1 progeny-arrested *rps-23(0)* larvae. The percentage of larvae with divided M cells was counted. This experiment was conducted across 9 biological replicates, with at least 15 arrested homozygous larvae assessed in each replicate.

## Epidermis-specific RNAi by dsRNA soaking

We performed epidermis-specific RNAi by soaking hatched L1 larvae with dsRNA. To generate the dsRNA, we amplified the following genes (*unc-31*, *cpr-4*, *lbp-1*, *lbp-2*, *far-1*, *npa-1*, *cpr-1*, *mec-5*, *ttr-5*, *ttr-2*, *rab-3*, *ida-1*, *ric-19*, *unc-108*) from cDNA, and *wrmScarlet* from plasmid pGLOW39. *T7* promoters were added to both ends of the amplified fragments using appropriate primers. The dsRNA synthesis was performed using the MEGAscript T7 Transcription Kit (Invitrogen) following the manufacturer's instructions. The epidermis-specific RNAi strain used in this study have a background of inducible epidermal ribosome biogenesis (*degron*:: *GFP*::*rpoa-2; col-10p*::*TIR1; rde-1(ne300); rde-1(ne219); wrt-2p*::*rde-1*). Synchronized embryos were placed on NGM plates without bacteria to hatch. Subsequently, the L1 larvae were soaked in a solution of 1 µg/µl dsRNA for 24 hours. After soaking, they were transferred to NGM plates containing 1 mM IAA and incubated for 3 days at 20˚C.

## Sample and library preparation for RNA sequencing

Larvae with or without IAA treatment were collected in 50 mM NaCl and were cleaned from OP-50 bacteria by sedimentation through a 5% sucrose cushion including 50 mM NaCl. After sucrose cleanup of bacteria, larvae were flash frozen in 20 mM Tris–HCl (pH 7.4), 150 mM NaCl, 5 mM MgCl2 and ground in liquid nitrogen with mortars and pestles. The frozen worm powder was thawed on ice and mixed with 5 mM DTT, 1% Triton X-100, 100 µg/ml cycloheximide (Sigma Aldrich) and 5 U/ml Turbo DNase (Thermo Fisher Scientific). Around 1 ml TRIzol (Thermo Fisher Scientific) was added to the lysate, vortexed, and incubated 5 minutes at room temperature. To extract RNA, 200 µl volume of chloroform was added and then the

sample was mixed and spun at 15,000 rpm for 10 minutes. Aqueous layer was used for further RNA precipitation. Isolated RNA was isopropanol precipitated and 80% ethanol washed. Thermostable RNAseH (Lucigen) and a pool of 94 DNA oligonucleotides antisense to *C. elegans* ribosomal RNA were used to deplete rRNA from 100 ng total *C. elegans* RNA [74]. RNA-seq libraries were prepared using SMARTer Stranded RNA-Seq kit (Clontech). Initially, RNA was alkaline fragmented at 95°C for 4 minutes followed by the protocol optimized <10 ng RNA input. To amplify the sequences, 12 to 14 cycles of PCR were used. Library DNA was then purified using Agencourt AMPure XP beads (Beckman Coulter). The resulting libraries were quantified with Qubit dsDNA HS Assay Kit (Thermo Fisher Scientific) and sequenced on NovaSeq 6000 v1.5, SP flow cell (Illumina).

### RNA-seq data analysis

Adapter removal (Truseq HT adapters), genome mapping (WBcel235), and assignment to protein coding genes were accomplished by using NextFlow preprocessing pipeline, Riboflow [76]. The raw reads per gene were extracted from the output ribo file using RiboR [76]. These reads were then analyzed for significant differences with and without IAA using Deseq2 analysis [46]. Gene expression $\log_2$ fold changes and base mean values that are used in **Figs 3**, **6** and **7** were predicted by Deseq2. The RNA-seq analysis values as well as raw reads are provided in **S1** and **S2 Tables**.

The RNA-seq libraries from this study can be accessed via the NCBI GEO database using the accession code GSE213367. The data are available at this link: https://www.ncbi.nlm.nih. gov/geo/query/acc.cgi?acc=GSE213367

For the GO term analysis, we took the significantly overexpressed or underexpressed genes and used FuncAssociate to analyze them. We entered the gene list into FuncAssociate, considering all detected genes as the background [56]. All significant GO categories resulting from the RNA-seq and proteomic analyses are provided in **S1, S2, and S3 Tables**. As some GO categories may overlap or encompass each other, we have selected representative GO categories that are significant for the respective plots in **Figs 5C** and **6D**.

We analyzed RNA-seq data using 5 different studies identifying specific gene sets. For the RNA-seq comparisons in **Fig 3**, we used 2 studies to identify specific gene sets. We previously analyzed L1 starved RNA-seq dataset alongside with RPL null animals (*rpl-5 (0)* and *rpl-33(0)*) to generate a significant gene set responding to ribosome large subunit deficiency or starvation [25,48]. We then compared these sets to the genes influenced by RPOA-2 depletion, calculating log-odds ratios and statistical significance using Fisher's exact test (**Fig 3C and 3D**). For dauer conditions, we examined genes with varied expression in dauer animals from McElwee and colleagues' microarray analysis [49], contrasting them with genes impacted by RNA Pol I depletion, applying Fisher's exact test for statistical analysis (**Fig 3E**). In **S5 Fig**'s RNA-seq comparisons, we used Kumar and colleagues' and Mueller and colleagues' studies for gene set identification. We compared DAF-16 target genes through ChIP-seq data and genes activated under low IIS in *daf-2(0)* animals according to their RNA-seq analysis [50] with genes influenced by RPOA-2 depletion, using Fisher's exact test for statistics (**S5C Fig**). We also compared significant gene list in response to UV irradiation obtained by microarray analysis [51] with the genes affected by RPOA-2 depletion, conducting Fisher's exact test for the comparison (**S5E Fig**). These analyses helped determine associations between the gene sets and the changes following RPOA-2 depletion, using log-odds ratios and *P* value to evaluate statistical significance.

### Western blotting

Animals with or without IAA treatment were collected and cleaned from OP-50 bacteria by sedimentation through a 5% sucrose cushion including 50 mM NaCl. The animals were then

flash frozen immediately in liquid nitrogen. The same volume of SDS loading buffer was added, and the samples were bead-beated for 30 seconds and incubated on a hot block for 5 minutes. Whole-worm lysates were separated on a 4% to 12% Bis-Tris protein gel (Thermo Fisher Scientific) and blotted onto a PVDF membrane. Antibodies against GFP (Thermo Fisher Scientific, #MA5-15256) and Actin (MP Biomedicals, #8691001) were used at dilution of 1:2,000 and 1:500, respectively. HRP-conjugated secondary antibodies (Thermo Fisher Scientific, #31431) and ECL reagents (Thermo Fisher Scientific, #34094) were used for detection. To quantify western blots, TIFF images were recorded for each blot using a chemidoc system, converted to 8-bit grayscale using Fiji, and the integrated intensity of each GFP and Actin band was calculated by Fiji. The GFP band intensity was normalized by the corresponding Actin band intensity. Each normalized GFP band intensity was expressed as a percentage of the control (−IAA, **S2B Fig**).

## Mass spectrometry proteomics and analysis

Animals, with or without IAA treatment, were grown on large plates seeded with the OP50 strain of *E. coli*. Nonstarved animals were washed off with the 50 mM NaCl buffer and then sucrose floated to remove all contaminants. Subsequently, the animals were flash frozen at −80°C until ready for lysis/digestion. For each sample, 10 μg protein was loaded into a 4% to 12% Bis-Tris protein gel (Thermo Fisher Scientific) and sent for MS analysis at the University of Texas System Proteomics Network. Raw label-free quantification (LFQ) intensities were used for protein quantification using DEP (Differential Enrichment of Proteomics Analysis) Package, in Bioconductor, R (https://rdrr.io/bioc/DEP/man/DEP.html). DEP was used for variance normalization and statistical testing of differentially expressed proteins. The resulting predicted $\log_2$ fold changes were used for proteomics-related Figures (**Figs 6, S10, and S11**).

## Supporting information

**S1 Fig. Homologues of ribosome biogenesis factors in *C. elegans*.** **(A-C)** Amino acid sequence alignments of ribosome biogenesis factors from 3 different species, *Caenorhabditis elegans*, *Saccharomyces cerevisiae*, and *Homo sapiens*. Alignments were performed using UniProt align function. **(A)** RPOA-2 in *C. elegans* shows homology to yeast RPA135 and human POLR1B. **(B)** GRWD-1 encoded by *Y54H5A.1* in *C. elegans* is homologous to yeast RRB1 and human GRWD1. **(C)** TSR-2 encoded by *Y51H4A.15* in *C. elegans* is homologous to yeast Tsr2 and human TSR2. **(D-F)** Comparison of the identity of RPOA-2 **(D)**, GRWD-1 **(E)**, and TSR-2 **(F)** in *C. elegans* with their homologues from *S. cerevisiae* and *H. sapiens*.
(TIF)

**S2 Fig. Effective degradation of RPOA-2 by the AID system.** **(A)** DAPI staining of L4 stage animals expressing degron::GFP-integrated RPOA-2. RPOA-2 is enriched in the nucleoli. **(B)** L4 stage animals expressing degron::GFP-integrated RPOA-2 and TIR1 ubiquitously were treated with 1 mM IAA. Animals were collected and lysed at 4 time points, 0-hour (0), 3-hour (3 h), 6-hour (6 h), and 24-hour (24 h). Western blots were performed using antibodies against GFP and Actin. The relative RPOA-2 protein levels were quantified using Fiji software. The numbers above the gel lanes represent the relative protein level normalized to Actin. **(C-E)** RPOA-2, GRWD-1, and TSR-2 are necessary for ribosome biogenesis. Polysome profiles of *degron::GFP::rpoa-2; eft-3p::TIR1* **(C)**, *grwd-1::degron::GFP; eft-3p::TIR1* **(D)**, and *tsr-2::degron::GFP; eft-3p::TIR1* **(E)** strains treated with and without 1 mM IAA for 24 hours from the L4 stage. The depletion of RPOA-2, GRWD-1, or TSR-2 by the AID system caused a dramatic decrease of ribosomes and polysomes. The underlying data for (C-E) can be found in Tab I in

S1 Data.
(TIF)

**S3 Fig. Depletion of a ribosome biogenesis factor results in early larval stage quiescence.**
**(A)** Embryos of the *degron*::*GFP*::*rpoa-2; eft-3p*::*TIR1* strain were treated with and without 1
mM IAA, and body length was measured over a span of 3 days with 40 animals for each condition per day. *P* values were calculated using an independent *t* test and adjusted by Bonferroni
correction. **(B)** Basal degradation of RPOA-2, GRWD-1, and TSR-2 were detected by GFP
fluorescence intensity in strains expressing global TIR1 compared to those without TIR1
expression. Higher degradation of TSR-2 with TIR1 was observed compared to that of RPOA-
2 or GRWD-1. GFP fluorescence was measured from 12 L4 stage animals of degron::GFP-integrated RPOA-2, GRWD-1, or TSR-2 strains. **(C)** Vulva invariant cell lineage was not observed
in *rpoa-2(ok1970)* animals after 4 days from the L1 stage. Arrows indicate vulva invariant cell
lineage in wild type. The underlying data for (A, B) can be found in Tab J in S1 Data.
(TIF)

**S4 Fig. Reversibility of early larval growth quiescence induced by inhibiting new ribosome
biogenesis. (A)** L4 stage animals were treated with (1 mM) and without (−) IAA for 24 hours.
Animals were immobilized on slides using 1 mM levamisole. All these animals grew to gravid
adults. Scale bar, 50 μm. **(B)** Growth reversibility was tested by treating embryos of *degron*::
*GFP*::*rpoa-2; eft-3p*::*TIR1* with 10 μM and 25 μm IAA from 1 to 5 days (x-axis) and then transferring them to plates without IAA. The presence of gravid adults and F1 progeny on plates
was inspected daily and the number of days taken to reach fertile adulthood was recorded (y-axis). No bar indicates that no gravid adults or F1 progeny were observed after removal of
IAA. The underlying data for (B) can be found in Tab K in S1 Data.
(TIF)

**S5 Fig. Gene expression signatures in response to global RPOA-2 depletion. (A)** Three representative significant GO categories with respective genes $\log_2$ fold changes were plotted. The
light blue, green, and magenta points indicate ribosome, protein synthesis, and chromatin/
transcription-related genes, respectively. **(B)** Deseq2 $\log_2$ fold change values in response to
global RPOA-2 depletion were plotted for overexpressed (light red) and underexpressed (light
blue) DAF-16 target genes [50]. **(C)** Shared gene expression changes in response to RPOA-2
depletion by RNA-seq and DAF-16 target genes were shown in the Venn diagrams. **(D)**
Deseq2 $\log_2$ fold change values in response to global RPOA-2 depletion were plotted for overexpressed (light red) and underexpressed (light blue) UV response genes [51]. **(E)** Shared gene
expression changes in response to RPOA-2 depletion by RNA-seq and UV response genes
were shown in the Venn diagrams. The underlying data for (B-E) can be found in Tab L in S1
Data.
(TIF)

**S6 Fig. Reversibility of development quiescence induced by inhibition of new ribosome
biogenesis in the epidermis. (A)** GFP displayed the expression pattern of RPOA-2 and BFP
showed the TIR1 expression in epidermis driven by *col-10* promoter. After 1 mM IAA treatment for 24 hours, RPOA-2 was specifically depleted in the epidermis. **(B)** Synchronized
embryos of strains expressing degron::GFP-integrated RPOA-2 and TIR1 in specific tissues
were treated with and without 1 mM IAA for 3 days. Body length was measured using Fiji software. Data were obtained from 34 animals without IAA and 50 animals with IAA treatment
for each strain. **(C)** Embryos expressing a degron::GFP-integrated ribosome biogenesis factor
(RPOA-2, GRWD-1, or TSR-2) and TIR1 in epidermis (*col-10p*) were exposed to 10 μM IAA
for 3 days and then transferred on plates without IAA for another 3 days. The percentage of

animals that recovered back to gravid adults were measured ($n = 40$). Animals were immobilized by 1 mM levamisole. The underlying data for (B, C) can be found in Tab M in S1 Data.
(TIF)

**S7 Fig. The growth quiescence in response to epidermal ribosome biogenesis is *daf-16* and *daf-18* independent. (A)** Animals of *daf-18(ok480)* and *daf-16(mu86)* did not show larger growth compared to wild type when the epidermal ribosome biogenesis was inhibited (*grwd-1*::degron::GFP; *col-10p*::TIR1, +IAA). Data are expressed as body length measured from 3 independent experiments with at least 18 animals in each replicate. **(B)** *daf-16* or *daf-18* RNAi did not affect animal growth in the absence of new epidermal ribosomes. Synchronized embryos were grown on NGM with 1 mM IAA for 3 days. Data are expressed as body length measured from 40 worms. *P* values were calculated using an independent *t* test and adjusted by Bonferroni correction in **(A, B)**. **(C)** The vulval extracellular space (indicative of transition into L4 stage) was not observed in *daf-18(ok480)* and *daf-16(mu86)* mutants when epidermal ribosome biogenesis was inhibited from embryos for 5 days. **(D)** Representative images of strain *daf-16*::mKate2; *grwd-1*::degron::GFP; *col-10p*::TIR1 that were grown from embryos on NGM with and without 1 mM IAA for 24 hours. **(E)** When these animals (in **D**) were transferred to survival NGM without *E. coli* for 30 minutes, animals in both conditions showed nuclear localization of DAF-16::mKate2. Animals were immobilized on slides using 20 mM sodium azide. Arrows indicate nuclear DAF-16 localization. Scale bar, 10 μm. The underlying data for (A, B) can be found in Tab N in S1 Data.
(TIF)

**S8 Fig. Gene expression changes at the RNA level in response to the global and epidermal RPOA-2 depletion. (A)** Log$_2$ fold changes of coding genes (y-axis) in response to epidermal RPOA-2 depletion (*degron*::GFP::*rpoa-2*; *col-10p*::TIR1) were plotted with respect to control (*degron*::GFP::*rpoa-2*) (x-axis). Log$_2$ fold changes and base mean values were calculated using Deseq2. Genes showing more than 2-fold differential expression were marked in red (overexpressed) and blue (underexpressed), and symbols indicate genes that were differentially expressed at least 16-fold. **(B)** Spearman correlation across different replicates was plotted using a clustered heatmap. The underlying data can be found in the S1 Table.
(TIF)

**S9 Fig. Epidermal RPOA-2 depletion results in cell nonautonomous gene expression changes at the RNA level.** Single-cell t-SNE plots for **(A)** 2 selected underexpressed genes (*C14A11.2* and *C37C3.11*) and **(B)** 4 selected overexpressed genes (*app-1*, *T22B7.4*, *ZK1098.3*, and *dod-18*). The t-SNE plots were generated using L2 stage single-cell RNA-seq data and single-cell-worm RNA software [57]. The colored points from t-SNE plots were original; however, their size was enlarged to ease visualization. The underlying data can be found in Tab O in S1 Data.
(TIF)

**S10 Fig. Gene expression changes at the protein level in response to epidermal RPOA-2 depletion. (A)** Label-free intensity (LFQ) based mass spectrometry quantification of proteins in response to the epidermal RPOA-2 depletion using the DEP package. Proteins showing more than 2-fold over- and underexpression were marked in red and blue, respectively. **(B)** Summary of the cellular location and function of differentially expressed proteins in response to RPOA-2 depletion in the epidermis. **(C)** Bar charts showing the expression of cytoplasmic and mitochondrial ribosomal protein genes at the RNA (orange) and protein (purple) levels. The underlying data for (C) can be found in Tab P in S1 Data.
(TIF)

**S11 Fig. The *unc-31* expression pattern and level in L2 stage animals by single-cell RNA-seq. (A)** Respective TPM (transcripts per million) values *of unc-31* gene in different tissues were plotted using single-cell expression data from L2 animals [57]. **(B)** Sense and antisense DNA fragments targeting *unc-31* gene and control gene (*mKate2*) driven by *wrt-2* promoter were injected to the inducible epidermal ribosome biogenesis inhibition strain. Animals with reduced UNC-31 grew significantly larger compared to control when the epidermal ribosome biogenesis was inhibited (*degron*::*GFP*::*rpoa-2*; *col-10p*::*TIR1*, +IAA) from embryos for 3 days. Data were obtained from 3 independent experiments with 16 animals for each replicate. **(C)** Double-strand RNA targeting the *unc-31* gene and control gene (*wrmScarlet*) transcribed in vitro was used to soak L1 larvae of epidermis-specific RNAi and epidermis-specific inducible ribosome biogenesis strain for 24 hours. The soaked larvae were then transferred to NGM plates with IAA for 3 days. Reducing *unc-31* expression by soaking significantly increased worm body length compared to the control. Animals were immobilized using 0.5% 1-phenoxy-2-propanol. Each 5× image was analyzed by custom MATLAB script (S1 **Text**). Data were obtained from 3 independent experiments with 27 animals for each replicate. Statistical significance was determined using an independent *t* test. The underlying data for (B, C) can be found in Tab R in S1 Data.
(TIF)

**S12 Fig. Expression patterns of DCV components and epidermal cells. (A)** The expression patterns of a DCV component gene, *rab-3*, and epidermal nucleus. The *rab-3* promoter drove the expression of a fluorescent protein gene, *wrmScarlet*, indicating the expression pattern of the *rab-3* transcript. Blue fluorescent protein (BFP) fused with a nuclear localization signal (NLS) was expressed in the epidermal nucleus driven by the *col-10* promoter. **(B)** The expression patterns of *rab-3* and epidermal cells labeled by a cytoplasmic GFP (col-10p::GFP). **(C)** Localization of endogenous IDA-1 and epidermal cells labeled with cytoplasmic green fluorescence (col-10p::GFP) in live animals. The *wrmScarlet* fluorescent protein gene was inserted in the C-terminus of the endogenous *ida-1* gene. Magenta indicates the expression pattern of IDA-1, and green indicates epidermal cells. L3 to L4 stage animals were immobilized using 1 mM levamisole.
(TIF)

**S1 Table. RNA-seq analysis of gene expressions in response to global and epidermal RPOA-2 depletion.** First tab: Raw counts in tidy format. Second tab: RNA-seq analysis in response to global RPOA-2 depletion. Third tab: RNA-seq analysis in response to epidermis-specific RPOA-2 depletion. Fourth tab: Shared significant changes in response to both epidermal and global RPOA-2 depletion. Fifth tab: Shared significantly enriched GO categories and GO gene attributes in response to global and epidermal RPOA-2 depletion.
(XLSX)

**S2 Table. GO enrichment analysis of shared underexpressed genes among starvation, dauer response, and RPOA-2 depletion.** The results of Gene Ontology (GO) enrichment for the shared underexpressed genes between RPOA-2 and dauer (first tab) or starvation (third tab), along with corresponding gene annotation attributes (second and fifth tab), are provided. Additionally, the shared list of unexpressed genes between starvation and RPOA-2 depletion was included.
(XLSX)

**S3 Table. Mass spectrometry data analysis of gene expression changes in response to epidermal RPOA-2 depletion.** The provided data includes raw label-free quantification (LFQ)

data (first tab), DEP (differentially expressed protein) analysis results (second tab), merged RNA-seq and proteomics results (third tab), and significant Gene Ontology (GO) enrichment analysis, accompanied by a table of GO gene attributes (fourth tab).
(XLSX)

**S4 Table. Constructs used in this study.**
(DOCX)

**S5 Table. *C. elegans* strains used in this study.**
(DOCX)

**S6 Table. Oligos used in this study.**
(DOCX)

**S1 Text. A worm body length analysis toolbox supported by MATLAB, which could automatically measure worm body length.**
(M)

**S1 Data.** Data underlying reported results in Figs as follows: Tab (**A**) Fig 1C; Tab (**B**) Fig 2B, 2D; Tab (**C**) Fig 3B-3E; Tab (**D**) Fig 4A-4C; Tab (**E**) Fig 5C-D; Tab (**F**) Fig 6A, 6D-6F; Tab (**G**): Fig 7A-7F; Tab (**H**) Fig 8A, 8C-8D; Tab (**I**) S2C–S2E Fig; Tab (**J**) S3A and S3B Fig; Tab (**K**) S4B Fig; Tab (**L**) S5B–S5E Fig; Tab (**M**) S6B and S6C Fig; Tab (**N**) S7A and S7B Fig; Tab (**O**) S9 Fig; Tab (**P**) S10C Fig; Tab (**R**) S11B and S11C Fig.
(XLSX)

**S1 Raw Images. Original western and membrane images for S2B Fig.**
(PDF)

## Acknowledgments

We thank Can Cenik and Arlen Johnson for critical reading of the manuscript. We thank Sarinay Cenik lab members, Steve Vokes, Andrew Fire, Keiko Torii, John Wallingford, and Suzan Ruijtenberg for discussions and feedback. We thank Dickinson lab for sharing their plasmids used for Cas-9-mediated genome editing. Protein identification was provided by the UT Austin Center for Biomedical Research Support Biological Mass Spectrometry Facility (RRID: SCR_021728). Some strains were provided by the CGC, which is funded by NIH Office of Research Infrastructure Programs (P40 OD010440).

## Author Contributions

**Conceptualization:** Qiuxia Zhao, Elif Sarinay Cenik.

**Data curation:** Qiuxia Zhao, Rekha Rangan.

**Formal analysis:** Qiuxia Zhao, Rekha Rangan, Cem Özdemir, Elif Sarinay Cenik.

**Funding acquisition:** Elif Sarinay Cenik.

**Investigation:** Qiuxia Zhao, Rekha Rangan, Elif Sarinay Cenik.

**Methodology:** Qiuxia Zhao.

**Project administration:** Elif Sarinay Cenik.

**Software:** Shinuo Weng.

**Writing – original draft:** Qiuxia Zhao, Elif Sarinay Cenik.

**Writing – review & editing:** Qiuxia Zhao, Rekha Rangan, Elif Sarinay Cenik.

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
