## [Editor Report · Decision Letter 0]

14 Oct 2022

Dear Dr SARINAY CENIK, 

Thank you for submitting your manuscript entitled "Hypodermal ribosome synthesis inhibition induces a nutrition-uncoupled organism-wide growth quiescence in C. elegans" for consideration as a Research Article by PLOS Biology. Please accept my apologies for the delay in getting back to you as we consulted with an academic editor about your submission. 

Your manuscript has now been evaluated by the PLOS Biology editorial staff, as well as by an academic editor with relevant expertise, and I am writing to let you know that we would like to send your submission out for external peer review.

Once your full submission is complete, your paper will undergo a series of checks in preparation for peer review. After your manuscript has passed the checks it will be sent out for review. To provide the metadata for your submission, please Login to Editorial Manager (https://www.editorialmanager.com/pbiology) within two working days, i.e. by Oct 16 2022 11:59PM.

Kind regards,

Richard

Richard Hodge, PhD

Associate Editor, PLOS Biology

rhodge@plos.org

PLOS

---

## [Decision Letter · Decision Letter 1]

23 Nov 2022

Dear Elif,

Thank you for your patience while your manuscript "Hypodermal ribosome synthesis inhibition induces a nutrition-uncoupled organism-wide growth quiescence in C. elegans" was peer-reviewed at PLOS Biology. Please accept my apologies for the delays that you have experienced during the peer review process. Your manuscript has been evaluated by the PLOS Biology editors, an Academic Editor with relevant expertise, and by three independent reviewers.

The reviews are attached below. As you will see, the reviewers find your manuscript interesting and novel but raise overlapping concerns with the overall strength of the mechanistic insights into the role of unc-31 in signalling from the hypodermis and whether other dense core vesicle pathway genes play similar roles. The reviewers also ask for additional experiments to validate the inhibition of ribosome biogenesis in the model.

Based on the reviews, it is clear that a substantial amount of work would be required to meet the criteria for publication in PLOS Biology. However, given our and the reviewer interest in your study, we would be open to inviting a comprehensive revision of the study that thoroughly addresses all the reviewers' comments and provides additional mechanistic details for the dense core vesicle signalling from the hypodermis, that goes beyond the observations reported with the unc-31 RNAi experiments. Given the extent of revision that would be needed, we cannot make a decision about publication until we have seen the revised manuscript and your response to the reviewers' comments. Your revised manuscript would need to be seen by the reviewers again, but please note that we would not engage them unless their main concerns have been addressed.

We appreciate that these requests represent a great deal of extra work, and we are willing to relax our standard revision time to allow you 6 months to revise your study. Please email us (plosbiology@plos.org) if you have any questions or concerns, or envision needing a (short) extension.

**IMPORTANT - SUBMITTING YOUR REVISION**

*Resubmission Checklist*

*Published Peer Review*

*PLOS Data Policy*

*Blot and Gel Data Policy*

Sincerely,

Richard

Richard Hodge, PhD

Associate Editor, PLOS Biology

rhodge@plos.org

REVIEWS:

Reviewer #1: In this manuscript, Zhao and coworkers identified ribosome biogenesis inhibition as a novel cell non-autonomous growth checkpoint in C. elegans. They found that auxin-inducible degradation of the RNA Pol I subunit rpoa-2 or two chaperones for ribosomal proteins (rrb-1 and tsr-2) in whole animals leads to an L2 larval arrest and shows similar gene expression signatures as UV-exposure. Moreover, there are also partial overlaps with dauer and starvation-induced L1 arrest, as well as with DAF-16 target genes. Hypodermis-specific depletion of RPOA-2 phenocopies whole body depletion, suggesting that loss of ribosome biogenesis in the hypodermis elicits an organism-wide response. Finally, the authors found that RNAi-mediated knockdown of unc-31 in the whole body or specifically in the hypodermis partially rescued the growth quiescence upon hypodermal RPOA-2 depletion.

Overall, the study by Zhao and coworkers is highly interesting to a broad readership and, therefore, suitable for PLoS Biology. The methods are state-of-the-art, especially the AID system and the comprehensive bioinformatic analysis. The newly generated degron strains were convincingly validated and represent a valuable tool for the community. The manuscript is well written, and data are presented clearly in all Figures, Supplementary Figures, and Tables.

The major drawback of the study is that direct experimental proof for cross-talk from the hypodermis to other tissues is missing, apart from the modest rescue of growth arrest by unc-31 RNAi and indirect evidence from bioinformatics.

Please find my specific comments below.

Major concerns:

1.) The whole study is based on the assumption that RPOA-2 depletion by AID inhibits ribosome biogenesis in whole nematodes or, specifically, in the hypodermis. This should be experimentally verified by demonstrating that fewer ribosomes are present, e.g., by quantifying 25S and 18S rRNA levels.

2.) Ribosome biogenesis inhibition will likely cause a reduction of global protein synthesis. The authors should verify this, e.g., by radioactive or puromycin labeling. Moreover, a recent study introduced a novel protocol to analyze protein synthesis in a tissue-specific manner (https://doi.org/10.1016%2Fj.crmeth.2022.100203). The authors could utilize this method to test if the degradation of RPOA-2 in the hypodermis also affects protein synthesis in other tissues.

3.) To complete the validation of their experimental model, the authors should verify in degron::GFP animals (lacking TIR1) that IAA exposure alone does not cause a growth phenotype.

4.) Although the inhibition of ribosome biogenesis at different steps by the AID system is elegant and specific, it would still be interesting to see if chemical inhibition of Pol I transcription, e.g., by CX-5461 or Actinomycin D, causes a similar growth quiescence phenotype. 

5.) The conclusion that daf-16 is repressed upon RPOA-2 depletion should be experimentally verified.

6.) Why were the worms raised at 16°C for some experiments (e.g., Figure 2C) and 20°C for others (e.g., 4D)? Please confirm that the temperature only affects the timing but not the larval stage of growth arrest.

Minor comments:

1.) It is not clear from how many independent experiments the indicated numbers of animals are derived.

2.) T-tests without corrections for multiple comparisons are unsuitable for comparing more than two groups (e.g., Figures 4A-C). Please include a correction for multiple comparisons or perform ANOVA with post hoc tests instead.

3.) Please include more details on the description of the bioinformatic methods. Were p-value, FC, or base-mean cutoffs used for the GO-term analysis? For the bioinformatic comparisons with published datasets, were the raw data re-analyzed using the same pipeline as the new data generated here, or were the already available gene lists used for the comparisons? Please also specify the original method (RNA-seq or microarray) and which sample groups of the published data were used for the comparisons.

4.) The sentence "The growth checkpoint…" (p3 line 7) is duplicated in p3 line 13.

5.) The last paragraph of the introduction (p3 line 18) is highly redundant (almost literal) with the abstract and could be shortened.

6.) Figure 4D: It is unclear what "L1" at the top refers to. It might be useful for readers to include "L1", "L2" and "L3" in the corresponding figure panels, like in Figure 2C.

7.) The introduction and discussion sections still contain minor typos and language errors.

8.) p15 line 25ff and p16 line 8ff are redundant ("Why does the hypodermis affect…").

Reviewer #2: The manuscript submitted by Zhao et al. is a continuation of an interesting study published by Cenik et al. in Developmental Cell in 2019, wherein the authors showed that maternal ribosomes are sufficient to carry out all aspects of embryonic development and that only when specific contexts are met during postembryonic development do animals arrest in the absence of newly synthesized ribosomes. In addition to this important information, these authors carried out an interesting series of mosaic analyses that allowed them to conclude that growth was mediated through some non-autonomous regulation between cells in the growing larva.

This new work attempts to build on this interesting observation by trying to identify the tissues involved in this regulation and the molecules that might contribute to the growth arrest associated with a lack of ribosomes or ribosomal function.

The authors use AIDs to genetically dissect the contributions of various tissues in this process in addition to extensive transcriptome and proteomic analysis to compare the changes in the ribosome-compromised animals. Although extensive, the authors finally focus on the role of a well known gene product that is involved in neuronal secretion of vesicles, which they eventually show may be playing a role in the hypodermis in order to mediate the ribosome-compromised signal to the rest of the animal to block growth and ensure appropriate scaling of the organism. This aspect (how an animal scales growth) of the manuscript is quite novel but is not fully elaborated beyond the observed organismal growth arrest that is induced following growth compromise in the hypodermis due to ribosomal disfunction. 

Overall, the manuscript is reasonably well written, although there are sections that are confusing and the rationale is not entirely clear. On the other hand the experiments are technically well performed and the data are solid and well controlled. The interpretations are also reasonable and are based on appropriate statistical analysis. The manuscript includes enormous amounts of data that have little impact on the most interesting aspect of the work, that being the basis of the non-autonomous effect of tissues to ensure growth arrest and appropriate scaling. 

The final series of experiments with unc-31 take the investigators incrementally forward in understanding how this non-autonomous signal is mediated. However, one could have intuitively predicted, or at least test this possibility, prior to engaging in the multiple omics-style analyses that contribute little to the final conclusions that the authors make. They do tell us that metabolic compromise is not the same as compromise of protein synthesis, which seems obvious for anybody that works on metabolism, but comparing all these omic signatures with other states where starvation associated quiescence is involved does provide some novel information. Unfortunately, the few proteomic hits that encourage the authors to proceed toward examining a role for secreted factors are mentioned, but not followed up or validated, making the entire contribution of these studies to the final conclusions pretty minor. 

I have listed a few concerns that I find problematic with the manuscript.

The authors claim that the cell non-autonomous signal originates in the ribosome-compromised hypodermis to regulate growth in non-compromised tissues. This may be overinterpeted since, as indicated in Figure 4, most tissues show some effect on limiting growth when they are compromised for ribosomal function. This could mean that all tissues are capable of this form of regulation to scale the organism, but because the hypodermis is one of the largest organs, its role is the most obvious. It is therefore possible that unc-31 may be playing a similar role in all the other tissues, but their individual contributions may simply more difficult to quantify since it would be inherently less than that of the comparatively much larger hypodermis. This might also be the basis of the difference that is noted between a loss of ribosome function in the hypodermis which results in an L3 arrest vs total loss of ribosomal function which causes an L2 arrest.

These data incrementally advance what the correpsonding author demonstrated in 2019 with a Dev Cell paper. The key question that I assume they are interested in answering is how this cell non-autonomy is generated. If the authors performed a genetic screen to identify unc-31 as a suppressor of the M cell division regulation, they must have also found the secreted target that is released by UNC-31 to mediate the regulation. This information is critical and would embellish this current work/submission. 

The limited effect of the unc-31 RNAi on suppressing organismal growth when eliminated in the hypodermis is accounted for by indicating that the RNAi might not be efficient or that there is redundancy. The unc-31 RNAi efficiency in the hypodermis can be controlled to test this idea, but alternatively, it could be due to unc-31 function in other tissues that contribute to the non-autonomous growth regulation. This needs to be elaborated and/or more rigorously investigated. 

I am not sure that this regulation can be referred to as a checkpoint, since there is no obvious contingency that needs to be met to advance in the growth pathway. However, the idea of tissue scaling is intriguing and that tissues need to communicate in order to grow together and not in an uncoordinated manner. Maybe the focus of this work could be adjusted to address this developmental problem without invoking some growth checkpoint that does not have an obvious trigger.

The proteomic targets that are cherry-picked from the data include secreted proteins that the authors suggest are indicative of some secretory pathway being involved. The target ENDU-2 is indeed a secreted endoribonuclease that culls specific somatic RNAs from the germ cells. It is not clear what this protein would be doing in this context, but if it is indeed present in higher levels, then it should leave some kind of change in the steady state RNA footprint, which might be relevant for this phenomenon. However, the authors mention it as a means of corroborating their secretion idea, and do not do any further experiments to follow up on its involvement. What about CPR or TTR gene products? Why were these not validated for their implication in this process? This leads the reader to surmise then that they simply aren't involved, which they might not be, and therefore makes all the other conclusions about this proteomic analysis and its relationship to the authors' claims somewhat questionable.

I also noted number of minor issues which I have listed below:

Page 9 Lines 5/6- this is awkward. Seems to be missing "is" or should read "differs"

Page 15 Line 2- ; the mesoblast....and the vulval precursor cells...

Page 18 Line 13- r in rad should be italicized

Page 29 Line 10- The language here seems incorrect. UNC-31 does not promote growth in my opinion. It is passive and is necessary for food intake which in turn may affect regulators that promote growth. If you introduce UNC-31 into a tissue that normally does not grow it will not suddenly grow because UNC-31 is present. Therefore, it permits growth due to its role in neuromuscular control of foraging behaviours and maybe even pharyngeal pumping.

Page 29 The entire section pertaining to the neuronal investigation of unc-31 and its role in this phenomenon is a bit confusing to read. Setting out the rationale of these experiments in a clearer way would make the section easier to understand and interpret.

Reviewer #3: This manuscript uses tissue specific inhibition of C. elegans orthologs of ribosome assembly factors to show that hypodermal (epidermal) ribosome function is required for overall growth of the animal. This work follows up previous studies on genetic mosaics for ribosome synthesis that show a developmental arrest; it extends the analysis to inhibit ribosome assembly in specific tissues and examines the effects on global gene expression and proteomes. The authors find expression of many genes is altered in these conditions, among which are several secreted proteins. The authors hypothesize that epidermal ribosome inhibition results in activation of a secreted signal and find evidence for the dense core vesicle regulator unc-31/CAPS being involved in the non-autonomous quiescence phenotype. Overall this work extends our understanding of ribosome function in vivo. The experiments are solidly performed and the data sets should be useful to others in the field. The main concerns are that the mechanism by which dense core vesicle signaling might be activated is not explored in depth. Other concerns are minor and could be addressed by toning down some claims or revising the presentation.

Mechanism of unc-31 in epidermis

The authors find evidence dense core vesicle regulator unc-31 functions in the epidermis to mediate the quiescence response. This finding is slightly unexpected as most prior work on unc-31 expression (e.g. a recent study on unc-31 knockins from Pocock lab) indicated it was primarily expressed in the nervous system. The current study could be strongly improved by direct evidence (beyond public RNAseq data) that unc-31 is expressed in the epidermis, or whether epidermis has dense core vesicles by cell biological or ultrastructural criteria. The latter could be checked using public domain datasets. The study would also be strengthened by test of whether other DCV pathway genes (e.g. ida-1, etc) play similar roles to unc-31. It is noted that the observed effects of unc-31 inhibition are weak e.g. Fig 7E shows maybe a 5% change in body length. A slightly stronger effect is seen in unc-31's suppression of a different mutant, rps-23 (Fig 7A) but almost no details are provided about this experiment, so it is unclear if it bolsters the model that unc-31 is required for signaling from the epidermis. While promoters used for RNAi or AID transgenes are somewhat tissue specific, they are not absolutely so, cf. the single cell RNA data on col-10 in Fig 5D. Thus weak effects could potentially result from targeting of transcripts in other tissues. Overall the conclusions that unc-31 acts in the epidermis should be either toned down, or substantiated by further experiments.

Presentation/wording

The authors show convincingly that the auxin-AID system is effective to deplete the worm versions of putative ribosome synthesis factors, but it is less clear if ribosome synthesis itself is inhibited (as stated in the title, and elsewhere) or if these genes have some other essential functions. There are two issues: first whether the worm orthologs have been investigated to assure that they act similarly to other systems, and second whether inhibition of these genes has the same consequences on ribosome populations. The authors observe overlap in effects on gene expression but without cell biological or biochemical analysis of ribosomes per se, it remains less clear what the effect of these gene inhibitions is in vivo. The nomenclature of these genes could also be clarified, e.g. rrb-1 is not currently annotated as such in the worm database, but a similar gene rrbs-1 has been annotated (not mentioned in this work).

The authors frame their findings as an example of tissue level coordination of growth analogous to the role of Dlip8 in Drosophila. However my understanding is in the fly system inhibition of ribosome function slows developmental timing allowing growth over a longer period to reach a normal size, whereas in the C elegans work the result is quiescence / developmental arrest in all cases. It is not clear if the authors have demonstrated 'tuning' of ribosome synthesis that allows modulation of growth rate as opposed to arrest. These distinctions could be clarified in the discussion

The statistical analyses should be checked by a statistical expert. In some places (e.g. Fig 4A) it is unclear if there is any correction for multiple comparisons. The use of different y axes for each data plot is confusing and potentially misleading; bar charts should be consistent with a 'zero' baseline. 

C. elegans jargon such as hypodermis could be replaced with 'epidermis' throughout, as it has has long been accepted that nematode hypodermis is equivalent to epidermis. Moreover 'hypodermis' means something different in human skin biology.

The work could be put in better context by citations of prior studies of possible signaling from the C. elegans hypodermis to other tissues e.g. Madaan et al 2020 on regulation of BMPs by collagens.

---

## [Decision Letter · Decision Letter 2]

13 Jul 2023

Dear Dr SARINAY CENIK,

Thank you for your patience while we considered your revised manuscript "Epidermal ribosome biogenesis inhibition induces a nutrition-uncoupled organism-wide growth quiescence in C. elegans" for publication as a Research Article at PLOS Biology. This revised version of your manuscript has been evaluated by the PLOS Biology editors, the Academic Editor and the original reviewers.

Based on the reviews, I am pleased to say that we are likely to accept this manuscript for publication, provided you satisfactorily address the remaining points raised by Reviewers #2 and #3. In response to Reviewer #2, we suggest that you enlist the services of a professional editing service or ask a colleague to proofread the manuscript text. In addition, we ask that you please make sure to address the following data and other policy-related requests that I have provided below (A-D):

(A) We would like to suggest the following modification to the title: 

“Inhibition of ribosome biogenesis in the epidermis is sufficient to trigger organism-wide growth quiescence independently of nutritional status in C. elegans"

(B) You may be aware of the PLOS Data Policy, which requires that all data be made available without restriction: http://journals.plos.org/plosbiology/s/data-availability. For more information, please also see this editorial: http://dx.doi.org/10.1371/journal.pbio.1001797

-Supplementary files (e.g., excel). Please ensure that all data files are uploaded as 'Supporting Information' and are invariably referred to (in the manuscript, figure legends, and the Description field when uploading your files) using the following format verbatim: S1 Data, S2 Data, etc. Multiple panels of a single or even several figures can be included as multiple sheets in one excel file that is saved using exactly the following convention: S1_Data.xlsx (using an underscore).

-Deposition in a publicly available repository. Please also provide the accession code or a reviewer link so that we may view your data before publication. 

Figure 1C, 2B, 2D, 3B, 4A-C, 5C-D, 6A, 6D-F, 7A-F, 8A, 8C-D, S2C-E, S3A-B, S4B, S6B-C, S7A-B, S9, S10C, S11B-C

(C) Please also ensure that each of the relevant figure legends in your manuscript include information on *WHERE THE UNDERLYING DATA CAN BE FOUND*, and ensure your supplemental data file/s has a legend.

(D) We require the original, uncropped and minimally adjusted images supporting all blot and gel results reported in the following Figures:

Figure S2B

We will require these files before a manuscript can be accepted so please prepare and upload them now. Please carefully read our guidelines for how to prepare and upload this data: https://journals.plos.org/plosbiology/s/figures#loc-blot-and-gel-reporting-requirements

We expect to receive your revised manuscript within two weeks. 

*Published Peer Review History*

*Press*

Sincerely,

Richard

Richard Hodge, PhD

rhodge@plos.org

Reviewer remarks:

Reviewer #1 (Markus Schosserer, signs review): The authors addressed all previous concerns thoroughly. Thus, I recommend the publication of the manuscript in its current form.

Reviewer #2: The revised manuscript by Zhao et al. has addressed most of my queries as best as they can, while I believe they also adequately answered most of the questions and addressed most of the comments raised by the other reviewers as well. I did note that reading through the revised version that there were a number of careless errors and grammatical and editorial issues that were pointed out previously that were overlooked for whatever reason. One final proofread before resubmitting the new version would have really helped the manuscript. Nevertheless, there are still a number of grammatical and language errors that should be fixed by having the paper carefully proofread by a colleague to catch all the small (and not so small!) errors. I have noted some of the issues that I noted, although this list is not comprehensive.

Page 10-Lines 7-9 and 13-15 are identical!

Page 10-Line 24...is is...

Page 10-Line 25 a bypass mutation suppresses. This seems redundant.

Page 11-Line 19-"the" majority

Page 31/32-Lines 25-30 are completely redundant/almost identical to Lines 8-13 on page 32

Page 32-Lines 26-32 should probably be rewritten. First, it is not at all clear what the authors are trying to convey by the statement that unc-31 alleviates the growth quiescence. My understanding from the data is that the quiescence that is observed by reducing ribosome function in the hypodermis results in an unc-31-dependent growth quiescence through its ability to effect a non-autonomous effect on other tissues. In this case unc-31 doesn't alleviate the growth quiescence but rather mediates it. The mutation suppresses this effect. As for the rest, Line 32...differences....are not "is"

Page 46-Line 6 "Inhibition" is capitalized-Why?

also, throughout- germline (adjective) germ line (noun); wild-type (adjective) (the) wild type (noun); RNAseq or RNA-seq...try to be consistent

Reviewer #3: The authors have addressed my concerns and the manuscript is much improved and suitable for publication. However I would strongly advise not using the term "tuning" as the experiments all appear to be simple binary inhibitions of gene activity i.e. (unless I have missed it) there is no demonstration of a graded inhibition of ribosome biogenesis that would support the claim of "tuning".

---

## [Editor Report · Decision Letter 3]

26 Jul 2023

Dear Dr SARINAY CENIK,

On behalf of my colleagues and the Academic Editor, Sean Curran, I am pleased to say that we can accept your manuscript for publication, provided you address any remaining formatting and reporting issues. These will be detailed in an email you should receive within 2-3 business days from our colleagues in the journal operations team; no action is required from you until then. Please note that we will not be able to formally accept your manuscript and schedule it for publication until you have completed any requested changes.

PRESS

Best wishes, 

Richard

Richard Hodge, PhD

rhodge@plos.org

PLOS
